# N-formyl-stabilizing quasi-catalytic species afford rapid and selective solvent-free amination of biomass-derived feedstocks

Hu Li [1,2,3], Haixin Guo[2], Yaqiong Su[4], Yuya Hiraga[3], Zhen Fang [1], Emiel J.M. Hensen [4], Masaru Watanabe[2,3] & Richard Lee Smith Jr. [2,3]

Nitrogen-containing compounds, especially primary amines, are vital building blocks in nature and industry. Herein, a protocol is developed that shows in situ formed N-formyl quasi-catalytic species afford highly selective synthesis of formamides or amines with controllable levels from a variety of aldehyde- and ketone-derived platform chemical substrates under solvent-free conditions. Up to 99% yields of mono-substituted formamides are obtained in 3 min. The C-N bond formation and N-formyl species are prevalent in the cascade reaction sequence. Kinetic and isotope labeling experiments explicitly demonstrate that the C-N bond is activated for subsequent hydrogenation, in which formic acid acts as acid catalyst, hydrogen donor and as N-formyl species source that stabilize amine intermediates elucidated with density functional theory. The protocol provides access to imides from aldehydes, ketones, carboxylic acids, and mixed-substrates, requires no special catalysts, solvents or techniques and provides new avenues for amination chemistry.

[1] Biomass Group, College of Engineering, Nanjing Agricultural University, 40 Dianjiangtai Road, 210031 Nanjing, Jiangsu, China. [2] Graduate School of Environmental Studies, Tohoku University, 6-6-11, Aoba, Aramaki, Aoba-ku, Sendai 980-8579, Japan. [3] Research Center of Supercritical Fluid Technology, Graduate School of Engineering, Tohoku University, 6-6-11, Aoba, Aramaki, Aoba-ku, Sendai 980-8579, Japan. [4] Department of Chemical Engineering and Chemistry, Laboratory of Inorganic Materials Chemistry, Schuit Institute of Catalysis, Eindhoven University of Technology, P.O. Box 513, 5600 MB Eindhoven, The Netherlands. Correspondence and requests for materials should be addressed to Z.F. (email: zhenfang@njau.edu.cn) or to M.W. (email: meijin@scf.che.tohoku.ac.jp) or to R.L.S.Jr. (email: smith@scf.che.tohoku.ac.jp)

Fragile ecosystems of our Earth and societal concerns over climate change are motivating industry to shift chemical production from conventional fossil resources to renewable biomass[1–3]. As an important class of organic molecules, amines are widely applied in daily necessities and industrial commodities, such as fine chemicals, drugs, polymers, and functional materials[4–10]. Particularly, nitrogen species are contained in over 80% of the top 200 pharmaceuticals[11]. A number of bio-based amines (e.g., aryl-amines, furyl-amines, alkyl-amines, and alkanolamines) have been reported to be efficiently synthesized from small platform molecules via a range of reaction routes like reductive amination of carbonyl compounds[12–16], reductive aminolysis of simple sugars[17–20], and hydrogenation-decarbonylation of amino acids[21]. Among these approaches, catalytic reductive amination of carboxides using molecular hydrogen is extensively performed over either transition (Raney Ni) or noble metals (e.g., Ru, Pd)[12,22]. Nevertheless, the synthesis of primary amines is often difficult to achieve in the desired selectivity due to simultaneous formation of secondary and tertiary amines with some of the issues being formation of inactive complexes from catalyst deactivation, use of high-pressure hydrogen gas or pungent and basic ammonia[11,23–26], which impede practical manufacture. Although gaseous or aqueous ammonia can be employed at moderate pressures, organic solvents (e.g., tetrahydrofuran, dioxane, and alcohols) along with additives are typically involved[11,23–26]. Therefore, development of effective and green catalytic processes for preparation of primary amines and relevant N-containing molecules, especially from biomass resources, is required for future sustainable society.

Formamide is a versatile chemical feedstock for pharmaceutical manufacture and is used in softeners for paper/fiber and as solvent for ionic compounds, resins and plasticizers[27,28]. Extensive application of formamide renders it to be commercially available in large quantities in which it is industrially produced by either carbonylation of ammonia or ammonolysis of alkyl formate[27]. Treatment of formic acid with ammonia to give ammonium formate, followed by heating can also yield formamide[29]. Formamide has a large dipole moment with similar solvation properties as water[30–32]. The relatively high dielectric constant of formamide (ca. 108) compared with that of water (ca. 80) make it an interesting reactant[33,34] for microwave-assisted and flow chemistry studies. In the past decade, microwave has been explored for accelerating biopolymer transformations[35–38], while little attention has been given to upgrading small biomass-derived platform molecules.

Aldehydes, ketones and carboxylic acids in diverse structures are available from lignocellulosic biomass via thermal or catalytic processes, while those carbonyl compounds, especially furanic aldehydes are seldom studied for amination due to the easy furan-ring opening reaction and the active –CHO randomly vulnerable to amino species[23]. In previous reports, the use of appropriate metal catalysts is a prerequisite for efficient reductive amination, typically via two reaction routes, including initial reduction to alcohols followed by amination or first amination to give imines coupled with subsequent hydrogenation (Fig. 1)[39–41]. Leuckart-type reactions are a classical approach to produce N-formylated compounds in moderate selectivity under conventional heating conditions, but due to uncontrollable amination levels, and the employed substrates, they are typically limited to petroleum-based and relatively stable aldehydes or ketones, apart from necessary long reaction times[42,43]. The obtained products are generally restricted to bear $C_1$ or multiple common substituents, and the involved reaction mechanism is ambiguous[44]. Herein, we report an efficient solvent-free protocol for selectively controlling different levels of amination of pentose-derivable furfural (FUR) using formamide (AM) and formic acid (FA) or ammonium

formate (AMF) reactants under rapid heating conditions offered by microwave irradiation (Fig. 1). Presence of the formyl species (N-formyl carbinolamine, N-formyl imine, and N,N'-(furan-2-ylmethylene)diformamide (FDFAM)), originating from AM/FA/AMF reactants, plays key roles in forming nitrogen-containing products at specific levels. The protocol can be extended to series of other bio-based carboxides including sugars to give moderate to nearly quantitative yields of primary amines with equivalent methyl formate (HCOOMe) by post-treatment mild methanolysis. The catalytic system is applicable to efficient synthesis of N-formyl imides for producing combination compounds, polymer-free monomers or heterocyclic compounds from aldehydes, ketones, carboxylic acids, or their mixtures (Fig. 1). We elucidate reaction pathways of in situ generated N-formyl species with kinetic and control experiments, deuterium isotope labeling reactions, and density functional theory (DFT) calculations.

## Results

**Mono-substitution of N-formamide.** Our study began with the microwave-assisted amination of furfural with formamide (Table 1) that shows product yields for several reaction conditions. [1]H and [13]C NMR spectra of major isolated formamides **1** and **2** are provided (Supplementary Figs. 1 and 2). In absence of FA, only 3% yield of N,N'-(furan-2-ylmethylene)diformamide (FDFAM) with <5% FUR conversion was obtained at 160 °C in 10 min (Table 1, entry 1). Elevation of reaction temperature to 180 °C or use of a solid acid catalyst (Amberlyst-15) gave an increase in FDFAM yield (Supplementary Table 1, entries 1–2), but neither **1** nor **2** was formed due to the lack of a hydrogen source. Instead, addition of FA significantly promoted the initial amination reaction, exclusively giving secondary amide **1** in 92% yield (Table 1, entry 2) without high extent of further amination with FUR to tertiary amide **2** (only 5% yield) at complete FUR conversion. This implies that FA is an excellent H-donor apart from acting as an acidic species in the cascade reaction. To elucidate whether diformamide FDFAM is a byproduct or intermediate for formation of **1**, a time-course study at different reaction temperatures from 120 °C to 200 °C was conducted (Supplementary Fig. 3). At 120 °C, moderate FDFAM yields of 65% and 67% were obtained after 10 min and 15 min, respectively (Table 1, entry 3). Although FUR was not completely converted even after 30 min at 120 °C (Supplementary Fig. 3a), FDFAM was gradually consumed to afford **1** (up to 32%). When reaction temperature was raised to 140 °C, no more than 50% yield of FDFAM with a higher yield of product **1** (65%) was observed after 15 min (Supplementary Fig. 3b), while after reaching the maximum value (92%) at 160 °C, a decrease in the yield of **1** was detected by prolonging reaction time from 10 min to 15 min (Supplementary Fig. 3c). These results (Supplementary Fig. 3) show that the amination is a temperature-dependent process with FDFAM as a key intermediate, which is in accord with the modeled reaction rate constants (Supplementary Table 2). At 180 °C and 200 °C, complete FUR conversion occurred in 3 min and 2 min, respectively (Table 1, entry 4). However, tertiary amide **2** was more prone to be formed instead of secondary amide **1** especially for extended reaction times (Supplementary Fig. 3d, e). Notably, the mass balance (carbon basis) for all tested reaction systems was no less than 94% and the co-products were mainly FDFAM and tertiary amide **2** besides product **1** (Table 1, entries 1–4; Supplementary Fig. 3), and tertiary amine **4** with some unknown products being generated at 200 °C after long reaction times (Supplementary Fig. 4). These results hint that the presence of the N-formyl group, originating from either formamide or formic acid, can stabilize **1** or **2** at relatively harsh conditions, which is well-supported by the lower reactivity that **1** displays in

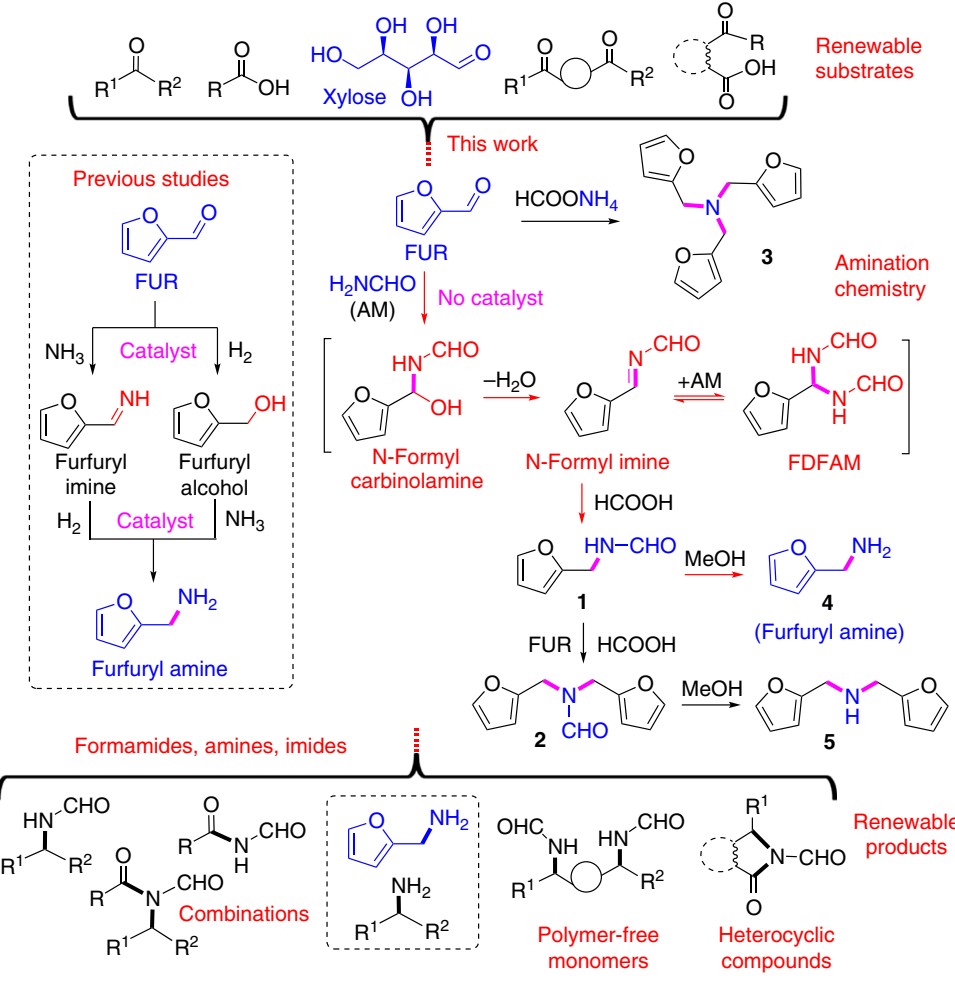

**Fig. 1** Schematic of selective production of amines, formamides and N-formylimides. FUR: furfural, FDFAM: N,N'-(furan-2-ylmethylene)diformamide, **1**: N-(furan-2-ylmethyl)formamide, **2**: N,N-bis(furan-2-ylmethyl)formamide, **3**: tris(furan-2-ylmethyl)amine, **4**: furfurylamine, **5**: bis(furan-2-ylmethyl)amine

**Table 1 Microwave-assisted amination of furfural (FUR)**

| Entry | Reactant (mmol) | | | | Temp. (°C) | Time (min) | FUR Conv. (%) | Product Yield (%) | | | | CB (%) |
|---|---|---|---|---|---|---|---|---|---|---|---|---|
| | FUR | FA | AM | AMF | | | | FDFAM | 1 | 2 | 3 | |
| 1 | 2 | – | 10 | – | 160 | 10 | <5 | 3 | 0 | 0 | 0 | 99 |
| 2 | 2 | 6 | 10 | – | 160 | 10 | 100 | <1 | 92 | 5 | 0 | 98 |
| 3 | 2 | 6 | 10 | – | 120 | 10 | 81 | 65 | 9 | <2 | 0 | 95 |
| | | | | | | 15 | 92 | 67 | 17 | 2 | 0 | 94 |
| 4 | 2 | 6 | 10 | – | 180 | 3 | 99 | 2 | 78 | 17 | 0 | 98 |
| | | | | | 200 | 2 | 100 | <1 | 61 | 35 | 1 | 97 |
| 5[a] | 2 | 6 | 10 | – | 180 | 60 | 83 | <1 | 69 | 10 | 0 | 97 |
| 6 | 2 | 6 | 12 | – | 180 | 3 | 100 | <1 | 98 | <1 | 0 | >99 |
| 7 | 2 | 6 | 6 | – | 200 | 3 | 97 | <1 | 12 | 83 | 0 | 98 |
| 8 | 2 | – | – | 12 | 195 | 5 | 100 | 0 | <2 | 2 | 88 | 92 |
| 9[b] | 2 | – | – | 12 | 195 | 5 | 100 | 0 | 6 | 3 | 80 | 89 |
| | | | | | | 10 | 100 | 0 | 11 | 11 | 66 | 88 |

Reactants formic acid (FA), formamide (AM) and ammonium formate (AMF)
[a]Oil bath
[b]0.5 mmol water added to reactant solution (5 wt% H₂O)
FDFAM: N,N'-(Furan-2-ylmethylene)diformamide, **1**: N-(furan-2-ylmethyl)formamide, **2**: N,N'-bis(furan-2-ylmethyl)formamide, **3**: tris(furan-2-ylmethyl)amine, CB: carbon balance

formic acid - formamide mixtures over that in its neat state at conditions of 160 °C under microwave irradiation (Supplementary Fig. 5).

**Mode of heating**. Conventional oil-bath heating was examined for furfural amination with formamide under identical reaction conditions, while hours were required for complete conversion of

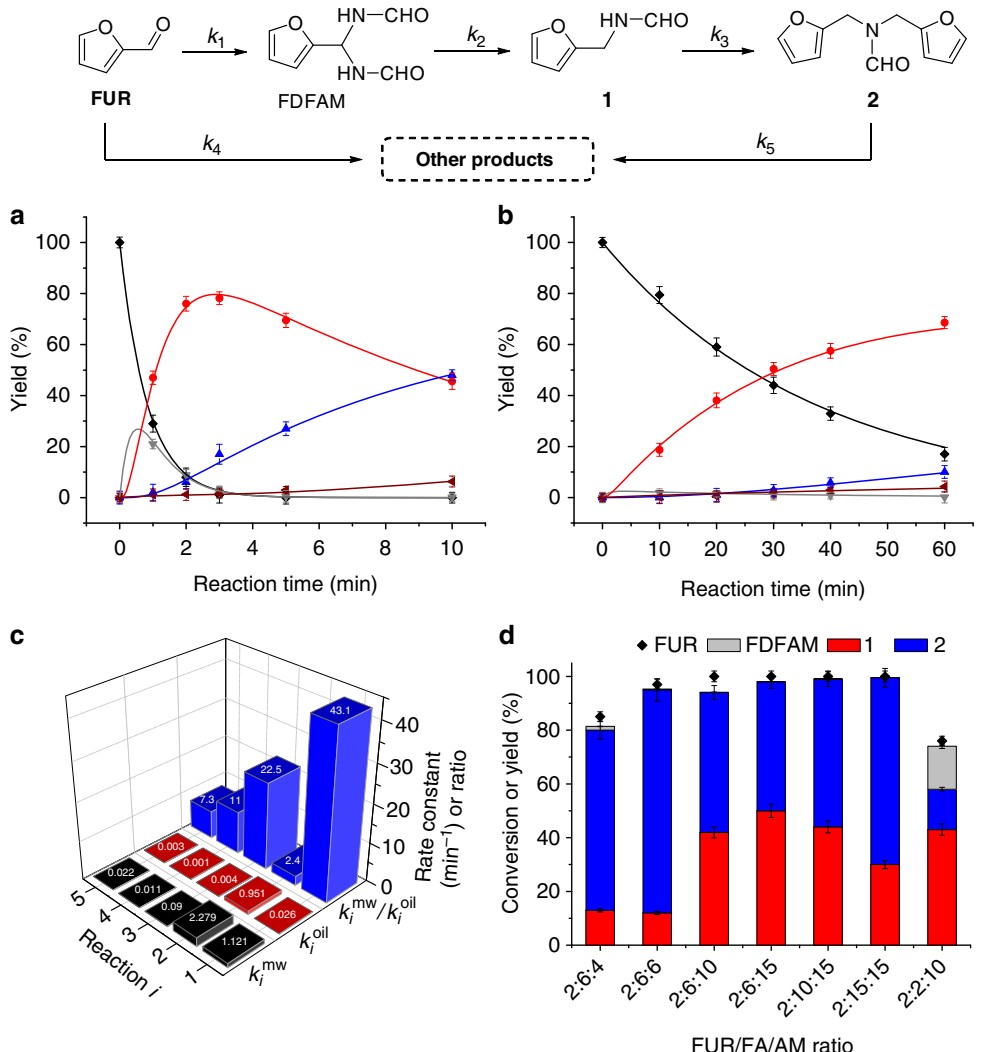

**Fig. 2** Amination of furfural (FUR) with formamide (AM). Time course for reactants heated by **a** microwave and **b** oil bath modeled with the assumption of pseudo-first-order reaction kinetics (black diamond, FUR; gray triangle, FDFAM; red circle, **1**; blue triangle, **2**; brown triangle, Other products). Rate constants of **c** microwave ($k_i^{mw}$) and oil heating ($k_i^{oil}$) and ratios ($k_i^{mw}/k_i^{oil}$) for cascade reactions ($i$) at reaction conditions (2 mmol FUR, 6 mmol FA, 10 mmol AM, 180 °C). Product distributions for **d** furfural/formic acid/formamide molar ratios with FUR set at reaction conditions (2 mmol FUR, 200 °C, 3 min). Error bars with standard deviation ($\sigma$) of ≤4.3%

furfural at reaction temperatures of (120 to 160) °C. A temperature of 180 °C was adopted for comparing product distribution and reaction rates of different heating modes (oil-bath and microwave). Based on the reaction pathways for amination of furfural with formamide and formic acid, rate data in Fig. 2a, b were fitted assuming pseudo first-order reaction kinetics, with rate constants ($k_1$–$k_5$) for each step being collected in Supplementary Table 2. Compared with microwave-assisted heating at 180 °C for complete conversion of furfural (3 min; Table 1, entry 4), only 83% furfural conversion was attained with oil-bath heating for 60 min (Table 1, entry 5). The ratio of kinetic rate-constants ($k_i^{mw}/k_i^{oil}$) were much larger for each reaction step as illustrated in Fig. 2c, which means that microwave facilitates C–N bond formation between furfural and formamide ($k_1^{mw}/k_1^{oil} = 43.1$) or product **1** ($k_3^{mw}/k_3^{oil} = 22.5$) that contains the amino group, while it has little effect in the reduction step ($k_2^{mw}/k_2^{oil} = 2.4$). The relatively high polarity of formamide estimated by its logP value (Supplementary Table 3) means that microwave energy most likely has a greater effect on formamide than formic acid. Microwave heating can promote side reactions derived from furfural ($k_4^{mw}/k_4^{oil} = 11$) and tertiary amide **3**

($k_5^{mw}/k_5^{oil} = 7.3$) as in Fig. 2c, however, these effects can be minimized with short reaction times (Fig. 2a, b), thus showing the great potential of selective production of either product **1** or **2** from furfural under microwave irradiation.

Temperature-time plots of reaction mixture and reactants clearly indicate the promotional role of formamide on heating rate (Supplementary Fig. 6). Although microwave heating effects are correlated with variable parameters (Supplementary Fig. 7), the heating ramp is controllable over a wide range[45], which is helpful for studying reaction selectivity with heating rate. In this regard, flow chemistry with the proposed protocol, as a rapid and regulatable heating approach could be a promising research topic for synthesis of nitrogen-containing compounds.

**Control of selectivity**. To demonstrate control of selectivity towards the secondary amide **1** or tertiary amide **2** from furfural amination, it was necessary to examine the FUR/FA/AM molar ratio. Gratifyingly, **1** was predominately formed in quantitative yields of 98% at 180 °C in 3 min (Table 1, entry 6), by slightly increasing molar content of AM from 10 mmol (entry 4) to 12

mmol (entry 6). The reaction process was monitored by recording GC-MS spectra of the liquid mixture (Supplementary Figs. 8–13) for different time intervals (up to 3 min), in which the high-purity of final solutions after reaction without post-treatment was confirmed by $^{13}C$ NMR (Supplementary Fig. 14). In previous literature[46,47], alkylation of primary amines can occur by removal of $NH_3$ to yield secondary or tertiary amines. In the present system, this type of reaction seems to take place when furfurylamine and FA were used as starting materials (Supplementary Fig. 15) giving **2** in yields of ca. 10% at 180 °C for 3 min reaction time as illustrated by GC-MS (Supplementary Fig. 16), and secondary amide **1** was formed as the primary product (ca. 85%) by N-formylation of furfurylamine with FA[48]. In contrast, almost no tertiary amide **2** was formed when secondary amide **1** was used as substrate with AM and FA (Supplementary Fig. 17), which means that the free amino group in furfurylamine is more vulnerable to react with itself and an active species (e.g., FA), as compared with **1** that contains the N-formyl group. To elucidate the activity of the N-formyl species in the FUR amination, product distributions were further studied by adjusting FUR/FA/AM molar ratios and using relatively high temperatures (Fig. 2d). Complete FUR conversion with approximately equivalent secondary amide **1** (42%) and tertiary amide **2** (52%) was observed at a FUR/FA/AM molar ratio of 2:6:10, while a decrease of AM to 4 mmol or FA to 2 mmol resulted in **2** (67%) or **1** (43%) being the main product, respectively (Supplementary Table 1, entries 3–5). Relatively lower FUR conversion (76%), but much higher FDFAM yield of 16%, was observed for a FUR/FA/AM ratio of 2:2:10, indicating that FA acts as both acid catalyst and H-donor in the cascade reactions. A relatively larger amount of both FA and AM was able to afford 100% FUR conversion with ca. 99% mass balance at FUR/FA/AM molar ratios of 2:10:15 and 2:15:15 (Fig. 2d). As expected, relatively high yields of tertiary amide **2** (83%) could be obtained by reducing the amount of AM (Table 1, entry 7), that probably decreased the amount of stabilized secondary amide **1** under microwave irradiation. The increased yield of **2** could also be due to the presence of relatively higher amounts of FA that allow FA to act as an acid catalyst to activate **1** for further amination. This speculation is supported by the significantly inferior reactivity between **1** and FUR without FA (<3% yield of **2**) or in presence of $Et_3N$ equivalent to FA (ca. 15% yield of **2**) under otherwise identical reaction conditions.

**Role of N-formyl species in intermediate stabilization**. Taking into account the effect that the N-formyl species has on product selectivity, ammonium formate (AMF), which is a promising ammonia precursor that forms in situ during thermal decomposition[49], was chosen for microwave-assisted amination of furfural, since AMF can donate hydrogen by formate[50]. Interestingly, the tertiary amine **3** was found to be the primary product with lesser amounts of the amides **1** (<5%) and **2** (<8%) at (185 to 205) °C (Supplementary Fig. 18), and a maximum yield of tertiary amine **3** (88%) was obtained at 195 °C in 5 min (Table 1, entry 8). Analysis of reaction mixtures with GC after different time intervals at 195 °C (Supplementary Fig. 19) confirmed that $NH_3$ was present in large quantities even after 30 min reaction time, while only low amounts of formamide were formed throughout the reaction period. The higher polarity of ammonium formate compared with that of formamide (Supplementary Table 3) may be positively correlated with energy adsorption efficiency under microwave irradiation and probably allows for rapid decomposition to generate $NH_3$, as compared with formation of formamide by dehydration. As a result, absence of primary and secondary amines, **4** and **5**, in the tested GC spectra was most likely the result of rapid transformation of these species

into tertiary amine **3** via two successive amination reactions (major) or into secondary and tertiary amides, **1** and **2** via N-formylation (minor) in the heating system (Supplementary Fig. 20). By increasing the molar ratio of FUR to AMF from 1:12 to 12:12, the yield of tertiary amine **3** was significantly reduced from 88% to 57% while both secondary and tertiary amides, **1** and **2**, were formed in almost constant yields (Supplementary Fig. 21). These unusual results highlight the significance of AMF on the promotion of three consecutive amination reaction processes to produce the tertiary amine **3** under microwave irradiation. We speculated that neither AM nor FA is formed to a large extent, which could be verified implicitly by additional experiments that showed enhanced selectivity toward secondary and tertiary amides **1** and **2** after addition of (5 to 25) wt% water that promoted formation of FA through hydrolysis of AMF (Table 1, entry 8 versus entry 9; Supplementary Figs. 22 and 23). Formation of tertiary amine **3** in high yield and selectivity can be attributed to the lack of N-formyl stabilizing species in the sequence of aminations. Presence of excess water would result in side reactions, such as furan-ring opening and condensation[51–53] to give unknown byproducts or insoluble humins, thus lowering the carbon balance.

**Reaction mechanism**. In the catalytic mechanism, two routes are typically considered for reductive amination of carbonyl compounds to amines (Supplementary Fig. 24), including initial reduction to alcohols followed by amination or first amination to give imines coupled with subsequent hydrogenation or hydrogenolysis[54]. In the present reaction system of FUR amination with AM and FA, gem-diamine (FDFAM), as discussed vide supra, was found to be the key intermediate that is accessible to secondary amide **1** after reduction with FA (Supplementary Fig. 25). However, neither imine nor carbinolamine intermediates were detected by GC-MS for tested reaction systems at temperatures up to 200 °C for variable reaction times (even <30 s). Considering the relatively low stability of these intermediates, $^{1}H$ and $^{13}C$ NMR, without high-temperature vaporization of samples during the analytical process, seemed to be a better tool for investigating reaction pathways. Several control experiments were designed, and the resulting spectra are shown in Supplementary Figs. 26–31. When FUR, AM, and FA were used as starting materials under reaction conditions of 180 °C for 3 min reaction time (Supplementary Fig. 26a), secondary amide **1** was found to be the major product with a certain amount of water (2.4 ppm) being formed with a fair amount (6.8 ppm) of AM residual. Similarly, only the sole secondary amide **1** was generated even in replacement of FUR and AM with furfuryl alcohol and furfurylamine, respectively (Supplementary Figs. 26d and 27), demonstrating the high reactivity between furfurylamine **4** and FA and the good stability of secondary amide **1** to resist attack of the active primary amino group in furfurylamine. In contrast, a mixture of the secondary amide **1** (major) and tertiary amide **2** (minor) were observed using FUR, furfurylamine, and FA as substrates (Supplementary Figs. 26c and 28–30), giving further evidence for the superior reactivity between furfurylamine and FA to that between furfurylamine and FUR, as well as confirming the pronounced promotional role of microwave on C–N bond formation rather than on the reduction step as shown previously in Fig. 2c. Moreover, furfuryl alcohol was stable in AM (Supplementary Fig. 26f), while it was degraded to levulinic acid in high degrees after addition of FA (Supplementary Fig. 26b), indicating the full capability of FA in the cleavage of C–O or C–N bond during reduction.

In this cascade reaction process, FA can be proposed to act as not only H-donor but also as an acidic catalyst. $^{1}H$–$^{13}C$ NMR

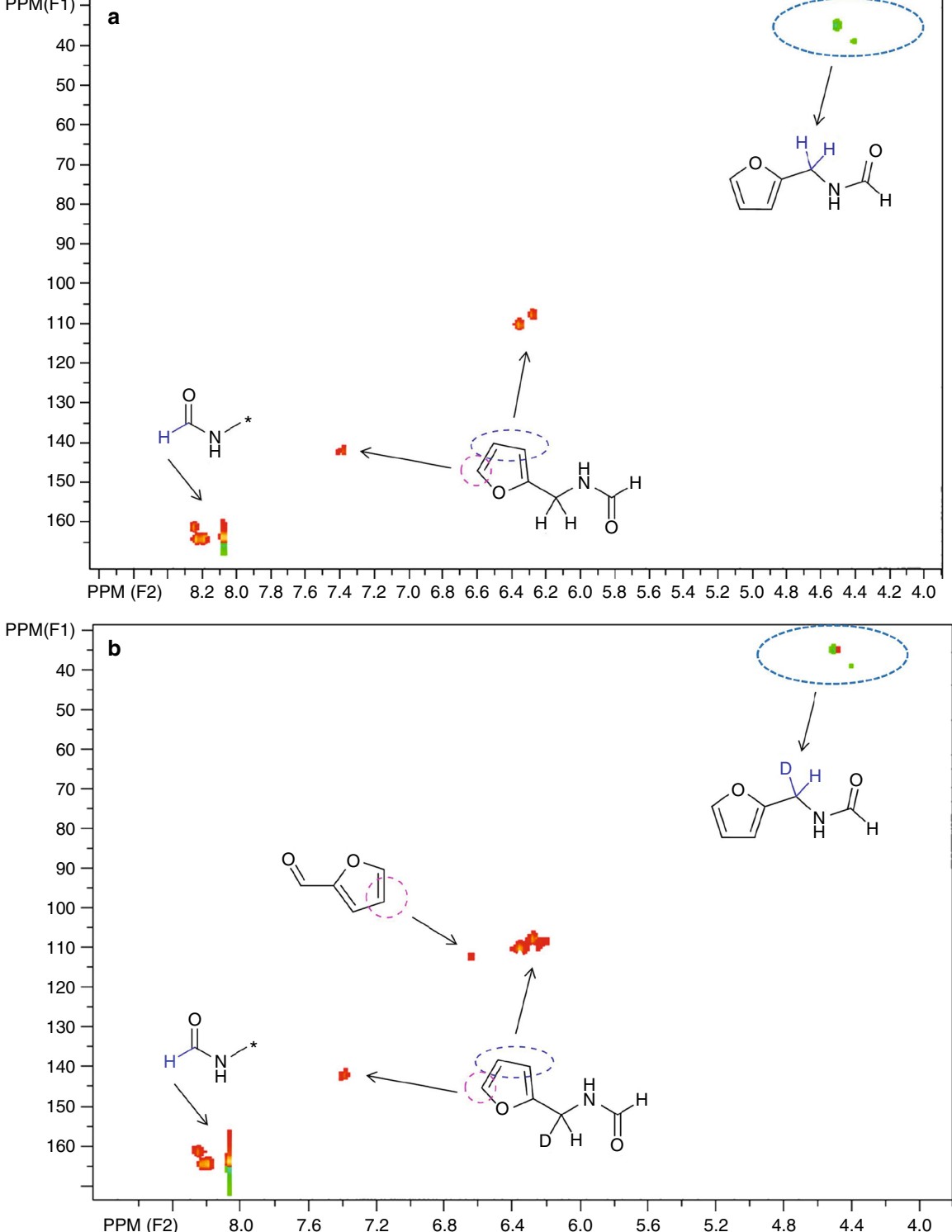

**Fig. 3** $^1$H-$^{13}$C NMR HSQC (DEPT) spectra of microwave-assisted amination of FUR and AM. Panel **a** normal FA and Panel **b** deuterium-labeled FA (red color: CH or CH$_3$, green color: CH$_2$); Reaction conditions: 2 mmol FUR, 6 mmol FA, 12 mmol AM, 180 °C, 3 min

HSQC (DEPT) spectra of FUR amination with AM using normal and deuterium-labeled FA (DCOOD) explicitly proved the incorporation of D from FA (Fig. 3, Supplementary Figs. 32 and 33), and moreover, the presence of unconverted FUR in the reaction system using deuterium-labeled FA (Fig. 3) indicated its relatively slower reaction rate compared with that using undeuterated FA. A primary kinetic isotope effect ($k_H/k_D = 3.8$) was observed, which is higher than typical transfer hydrogenation or dehydrogenation of FA[55,56], clearly affirming the additional role of FA possibly as an acid in the reaction process. Mass spectra showed that secondary amide **1** was formed with +1 and +2 mass shifts besides 125 ($m/z$, amu) in comparable intensity using DCOOD from that using HCOOH (Supplementary Figs. 34 and 35). This manifests that the N-formyl imine ($m/z + 2$) derived from carbinolamine ($m/z + 1$) along with FDFAM ($m/z + 1$) could all be intermediates in the formation of secondary

**Fig. 4** Computed free energy profiles for amination of FUR and AM using FA to product **1**. TS: transition state, IM: intermediate. Conditions and basis set: [$T = 180\,°C$, B3LYP/6-311+G(2s,2p)]. Values in parentheses are free energies (kJ mol$^{-1}$) based with respect to the starting energy of the three separated reactants FUR, AM, and FA

amide **1** (Supplementary Fig. 36). Based on the above results, despite being unable to directly trace the N-formyl imine or carbinolamine intermediate, the reaction pathways could thus be elucidated (Supplementary Fig. 25). With a relatively high thermal stability under microwave irradiation, FDFAM is rapidly produced in detectable amounts from in situ formed N-formyl imine or carbinolamine intermediates by addition of AM, and all of these species are further converted to secondary amide **1** through hydrogenation (for imine) or hydrogenolysis (for carbinolamine and FDFAM). With respect to further transformation of **1** to tertiary amide **2**, neither imine nor gem-diamine was formed as intermediate, except for the secondary carbinolamine (Supplementary Fig. 25). In this case, an appropriate relative ratio of furfural to AM for generation of the carbinolamine, as well as an optimal reaction temperature for its hydrogenolysis other than degradation by FA should be the key factor for selectivity control, which is in good agreement with the obtained results (Table 1, entries 6–7). In amination of furfural and AMF, a certain amount of AM was observed after reaction (Supplementary Figs. 26e and 31), showing the occurrence of AMF dehydration besides the dominant route to thermal decomposition of AMF into ammonia and FA (Supplementary Fig. 25). Due to limited amounts of FA forming in situ during the reaction, primary and secondary amines, **4** and **5**, seemed to be major intermediates that rapidly led to tertiary amine **3**. The nearly changeless yields of secondary and tertiary amides, **1** and **2**, at different reaction time courses (Supplementary Fig. 18) may be closely related to their high stability in AM, while it may possibly testify to the superior activity of FA in the reduction of imine to that of formylimine.

**DFT calculations**. Considering the significance of primary amines and relative simplicity of the product distribution, DFT calculations [$T = 180\,°C$, B3LYP/6-311+G(2s,2p)] were performed to study the reaction mechanism. In the reductive amination of FUR and NH$_3$ to furfuryl amine in the presence of metal catalyst, a furfurylimine (furan-2-ylmethanimine) has been proposed to be a key intermediate in the reaction mechanism, although it could not be detected[39,40]. However, in the absence of metal catalyst,

computational calculations disclose that high energy barriers for the transition state (**TS**, as high as 188 kJ mol$^{-1}$) exist in the initial formation of the furfurylimine from FUR and NH$_3$ (Supplementary Fig. 37). In sharp contrast, the activation energy for amination of FUR and AM in presence or absence of FA is relatively lower or higher (152 kJ mol$^{-1}$ or 245 kJ mol$^{-1}$; Supplementary Fig. 37), clearly demonstrating the promotional role of FA during the initial C-N bond formation and the stabilizing effect of N-formyl species in the reaction system, respectively.

Based on the obtained experimental results, gem-diamine FDFAM, N-formyl imine and N-formyl carbinolamine are key species in the quasi-catalytic amination pathway. To elucidate the underlying mechanism, different paths proceeding through these species were devised giving the obtained computational free energy profiles (Fig. 4, Supplementary Fig. 38). The target product was designated as N-(furan-2-ylmethyl)formamide (**1**) starting from FUR and AM with FA. Initially, C-N bond formation between FUR and AM occurs via direct addition to give corresponding N-formyl carbinolamine, and free energy of **TS1** as 152 kJ mol$^{-1}$, in agreement with the experimentally observed activation energy 148 kJ mol$^{-1}$ under oil-heating conditions (Supplementary Fig. 39). The slightly higher reaction barrier is consistent with the relatively high temperatures (e.g., 180 °C) that allow the reaction to proceed smoothly. With FA acting as an acid, the formed N-formyl carbinolamine can be further transformed into N-formyl imine by dehydration via **TS2** ($\Delta G^{\#} = 72$ kJ mol$^{-1}$, Fig. 4). When FA is absent for the same dehydration step, much higher activation energy (194 kJ mol$^{-1}$, Supplementary Fig. 37c) was required, explicitly clarifying the acidic role of FA. Afterwards, the formed imine is capable of being further converted to **1** using FA as H-donor sequentially overcoming **TS3** via **IM1**. In view of the relatively low reaction barrier ($\Delta G^{\#} = 62$ kJ mol$^{-1}$ versus 113 kJ mol$^{-1}$, Fig. 4), product **1** is preferably formed from N-formyl carbinolamine than from FDFAM via **TS5**. However, sole FDFAM with relatively low free energy (15 kJ mol$^{-1}$) generated in situ from N-formyl imine with AM and FA through **TS4** ($\Delta G^{\#} = 104$ kJ mol$^{-1}$) accounts for the observation of certain quantities of FDFAM during the initial

**Table 2 Selective production of formamides and primary amines from carboxides**

| Entry | Substrate | | Product yield (%) | |
|---|---|---|---|---|
| | R1 | R2 | Formamide [a] | Primary amine [b] |
| 1 | furan | H | 98 | 94 |
| 2 | methylfuran | H | 97 | 95 |
| 3 | phenyl | H | >99 | 97 |
| 4 | dimethoxyphenyl | H | 97 | 92 |
| 5 | styryl | H | 95 | 98 |
| 6 | HO-ethyl | H | 87 | 95 |
| 7 | propyl | H | >99 | 97 |
| 8 | isopropyl | CH3 | 85 | 96 |
| 9 | butylene (cyclic) | | 94 | 98 |
| 10 | pentylene (cyclic) | | 92 | 97 |
| 11 | phenyl | CH3 | 99 | >99 |

[a]Reaction conditions: 2 mmol substrate, 6 mmol FA, 12 mmol AM, 180 °C (microwave), 3 min
[b]Reaction conditions: 1 mmol formamide, 2 mL methanol, 3 mmol $Cs_2CO_3$, 60 °C (oil bath), 1 h; Co-product is the equivalent methyl formate

reaction stage consistent with the experimental results. In other words, N-formyl carbinolamine seems to be involved in the predominant reaction route, while FDFAM as a key intermediate most likely acts to sustain high carbon balances observed experimentally by stabilizing reaction species.

In the progress of the reaction, proton-transfer interactions between AM and FA may also contribute to the superior reactivity of the catalytic system (Supplementary Fig. 40); microwave effects were not considered in the DFT calculations. Initial C–N bond formation was observed to be the rate-determining reaction step (Fig. 4), while the medium reaction energy barrier ($152 \, kJ \, mol^{-1}$) could be overcome by performing the amination at relatively high temperatures, which is in accordance with the oil-heating experimental activation energy ($148 \, kJ \, mol^{-1}$). However, under microwave irradiation, the experimental activation energy was found to be *ca.* $50 \, kJ \, mol^{-1}$, clearly elaborating the promotional role of microwave heating in the facilitation of C–N bond formation. With addition of SiC powder into reaction mixtures of FUR, AM, and FA (Supplementary Table 4), the reaction rate can be further accelerated despite forming a certain amount of carbon, which also affirms the positive role of rapid heating in the reaction process. When an oil bath was used as the heating source, the sluggish heating and cooling steps might impede amination to effectively take place.

Nevertheless, the key FDFAM species could be clearly identified, thus confirming the amination chemistry illustrated in context.

**Extension of substrate scope**. Conventional preparation of primary amines from ammonia generally involves unexpected reaction pathways such that metallic materials with specific catalytic functionalities are required to control product selectivity[39,57,58]. Here, furfurylamine **4** was found to be synthesized in good yields (94%) by simple treatment of secondary amide **1** in methanol with $Cs_2CO_3$, and secondary amine **5** could be obtained in 96% yields from tertiary amide **2**. Encouraged by the outstanding selectivity shown in the previous trials, a number of biomass-derived carboxides were employed as starting materials to investigate the protocol generality, especially for selective production of primary amines (Table 2). As expected, all aldehydes and ketones examined could be efficiently converted into primary amines in mostly quantitative yields through the corresponding formamides (Table 2, entries 1–11). Although some aliphatic ketones or aldehydes containing active substituent groups (e.g., –OH) gave relatively inferior yields of formamides related to side reactions that occur under microwave irradiation (Table 2, entries 6, 8), the selectivity towards primary amines was still comparable with the others.

**Table 3 Synthesis of furfuryl amines from furfuryl aldehydes or sugars with formamides**

| Entry | Substrate | | | | Time (min) | Yield (%) |
|---|---|---|---|---|---|---|
| | Sugar | R | R$^1$ | R$^2$ | | |
| 1 | – | H | H | H | 3 | 92 |
| 2 | – | HOCH$_2$ | H | H | 2 | 63 |
| 3 | – | H | CH$_3$ | H | 4 | 90 |
| 4 | – | HOCH$_2$ | CH$_3$ | H | 3 | 61 |
| 5 | – | H | CH$_3$ | CH$_3$ | 5 | 85 |
| 6 | – | HOCH$_2$ | CH$_3$ | CH$_3$ | 3.5 | 56 |
| 7 | Xylose | H | H | H | 10 | 83 |
| 8 | Fructose | HOCH$_2$ | H | H | 6 | 74 |
| 9 | Glucose | HOCH$_2$ | H | H | 10 | 45 |
| 10 | Sucrose | HOCH$_2$ | H | H | 7.5 | 62 |
| 11 | Cellobiose | HOCH$_2$ | H | H | 10 | 26 |

Reaction conditions: 2 mmol furyl aldehyde or 1 mmol sugar, 12 mmol N-formyl compound, 6 mmol FA, 180 °C (microwave); When R$^1$ and/or R$^2$ = H, 2 mL methanol with 6 mmol Cs$_2$CO$_3$ was added after the reaction and microwave-treated at 120 °C for 4 min

**Fig. 5** Synthesis of N-formylimides and N-formamide moieties (**7**) from AM and carbonyl compound (**6**). Reaction conditions: 1 mmol **6**, 12 mmol AM, 12 mmol FA, 180 °C (microwave)

FUR and 5-hydroxymethylfurfural (HMF) are two important platform molecules derived from pentose and hexose saccharides, respectively[59,60], both of which could be subjected to amination (Table 3, entries 1 and 2). Besides formamide, *N*-methylformamide and *N*,*N*-dimethylformamide were able to be used as nitrogen sources, affording homologous *N*-methyl-furfuryl and *N*,*N*-dimethyl-furfuryl amines (Table 3, entries 4–6). However, presence of the hydroxy group in HMF, while being inert for amination, causes serious side reactions like condensation or polymerization to give insoluble black humins, especially after long reaction times.

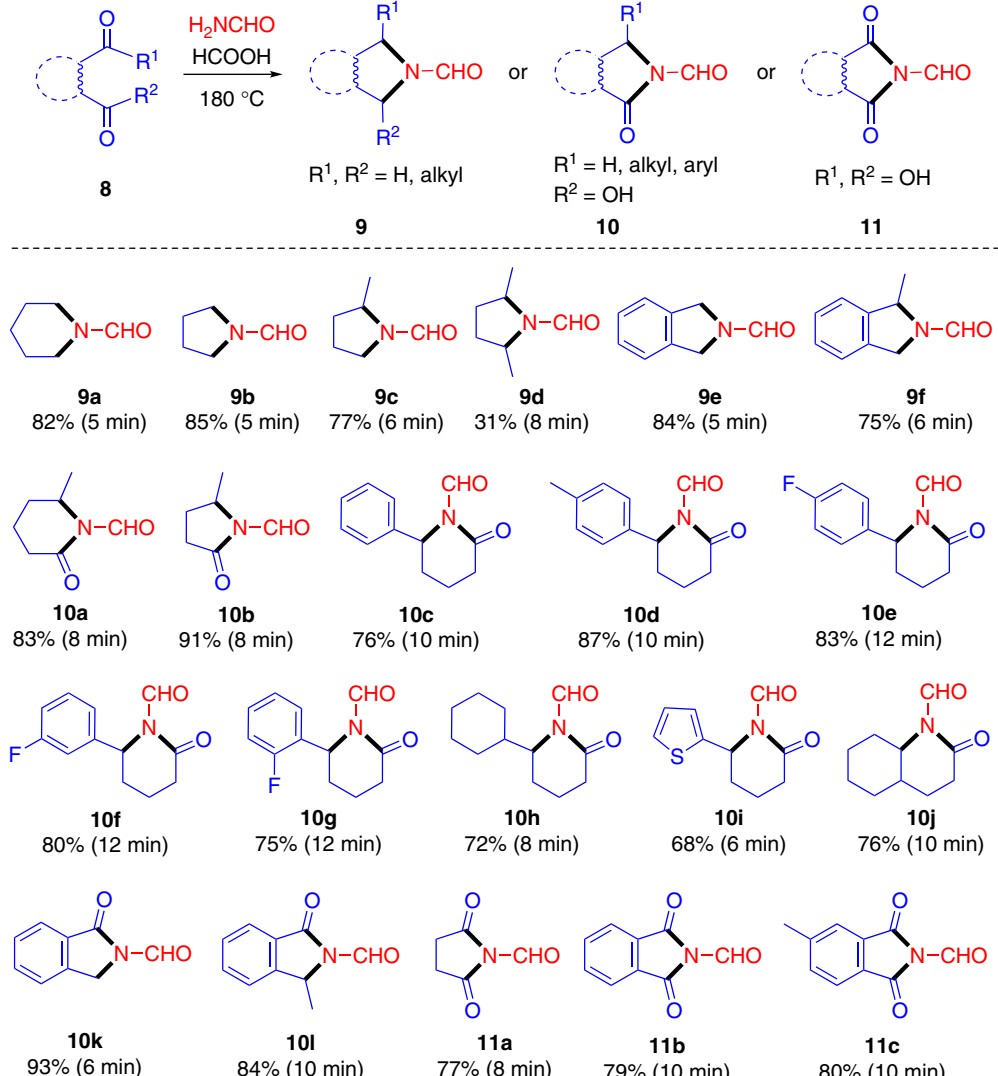

**Fig. 6** Synthesis of N-formyl cyclamines (**9**), N-formyl lactams (**10**), or N-formyl cyclic imides (**11**) from AM and dicarbonyl compounds (**8**). Reaction conditions: 1 mmol **8**, 8 mmol AM, 12 mmol FA, 180 °C (microwave)

Monosaccharides (xylose, fructose, and glucose) and disaccharides (sucrose and cellobiose) can be upgraded via the amination chemistry (Table 3, entries 7–10). Cellulose is not directly applicable to the amination chemistry, while its oxidized counterpart (dialdehyde cellulose) could be aminated to a great extent (Supplementary Figs. 41 and 42). These results demonstrate the feasibility and versatility of the quasi-catalytic amination chemistry for selective production of diverse nitrogen-containing compounds from renewable biomass resources.

N-formylimides are biologically active moieties and are versatile building blocks in organic synthesis and they are typically synthesized by N-formylation of amides or lactams employing sophisticated reagents as catalysts[61,62]. Herein, direct conversion of alkyl and aryl carboxylic acids **6** to corresponding N-formylimides **7** with moderate yields (61% to 84%) could be achieved using the quasi-catalytic amination chemistry within (4 to 7) min (Fig. 5). The reaction system was compatible with different dicarbonyl species, whereas carboxylic acid, aldehyde, and/or ketone groups were able to be simultaneously aminated into a wide range of nitrogen-containing compounds with N-formylimides and/or N-formamide moieties. Appropriate amounts of FA were found to be essential in obtaining optimal

product yields, otherwise the co-products would be formed by partial or complete removal of N-formyl species due to the presence of acidic carboxylic group in the substrates (Supplementary Fig. 43).

In the reaction systems (Fig. 5), neither cross-polymerized nor self-polymerized products were detected in high extents, which is ascribed to the stabilizing effect of the N-formyl species, as well as to steric hindrance effects (e.g., para-position for aromatic rings, trans-configuration for double bonds). Interestingly, both saturated linearly alkyl and ortho-substituted aromatic dicarbonyl compounds (**8**) are prone to undergo self-cyclization through two sequential amination processes after a relatively long reaction time (5–12 min), giving N-formyl cyclamines (**9**), N-formyl lactams (**10**), or N-formyl cyclic imides (**11**) in moderate to good yields (68–93%), which is closely dependent on the type of starting carbonyl species (Fig. 6). A poor yield of **9d** (31%) obtained from 2,5-hexanedione further indicated steric hindrance effects in the secondary amination. In this regard, an appropriate choice of substrates with desired structures and functionalities has great potential for selective synthesis of nitrogen-containing scaffolds.

Apart from the two-component amination between mono-carbonyl or dicarbonyl compounds and AM, a three-component

**Table 4 Three-component synthesis of N-formylimides from carboxylic acids, AM, and aldehydes/ketones**

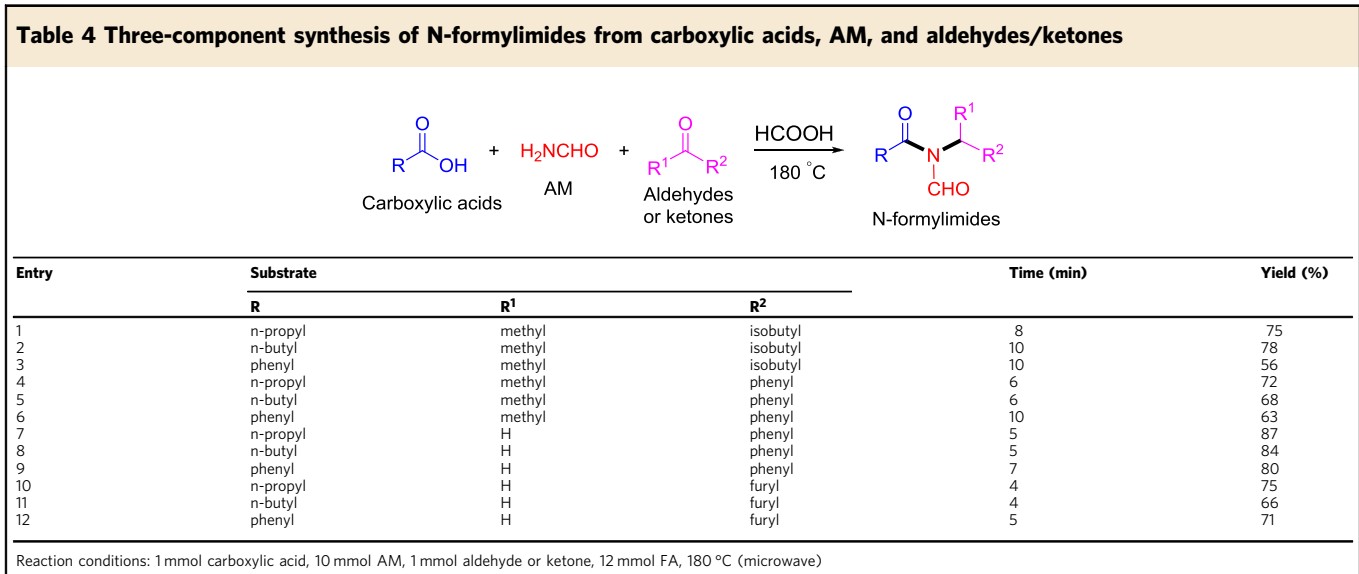

| Entry | Substrate | | | Time (min) | Yield (%) |
|---|---|---|---|---|---|
| | R | R¹ | R² | | |
| 1 | n-propyl | methyl | isobutyl | 8 | 75 |
| 2 | n-butyl | methyl | isobutyl | 10 | 78 |
| 3 | phenyl | methyl | isobutyl | 10 | 56 |
| 4 | n-propyl | methyl | phenyl | 6 | 72 |
| 5 | n-butyl | methyl | phenyl | 6 | 68 |
| 6 | phenyl | methyl | phenyl | 10 | 63 |
| 7 | n-propyl | H | phenyl | 5 | 87 |
| 8 | n-butyl | H | phenyl | 5 | 84 |
| 9 | phenyl | H | phenyl | 7 | 80 |
| 10 | n-propyl | H | furyl | 4 | 75 |
| 11 | n-butyl | H | furyl | 4 | 66 |
| 12 | phenyl | H | furyl | 5 | 71 |

Reaction conditions: 1 mmol carboxylic acid, 10 mmol AM, 1 mmol aldehyde or ketone, 12 mmol FA, 180 °C (microwave)

synthetic route was examined, where carboxylic acid, aldehyde/ketone, and AM were used as substrates (Table 4). Both alkyl and aromatic carboxylic acids are subjected to the three-component amination with AM and aldehydes or ketones, affording N-formylimides in moderate yields (56–87%). These results clearly demonstrate the versatility of the quasi-catalytic amination chemistry for controllable synthesis of formamides, amines, and imides.

## Discussion

We developed a simple two-step protocol for selective synthesis of primary amines that provides quantitative yields from many types of chemical and bio-related substrates. Not only primary amines, but also secondary and tertiary amides, tertiary amines and imides can be selectively and rapidly produced in several minutes under rapid heating with microwave irradiation by regulating reactant ratio, ammonia precursor, substrate type, and content or releasing mode of hydrogen species. The N-formyl quasi-catalytic species stabilizes formamides, thus affording superior selectivity toward secondary and tertiary amides, while the tertiary amine is most likely formed via reductive amination of furfural with ammonia formed in situ from thermal decomposition of ammonium formate where N-formyl species exists in relatively low amounts during reaction. Kinetic studies show that the C–N bond formation other than reduction step can be significantly enhanced under rapid heating with microwave irradiation compared with conventional convective heating. DFT calculations confirm that the initial C–N bond formation has a relatively high energy barrier in the rate-determining reaction step. Formic acid acts as both acid catalyst and H-donor, which promotes dehydration, selective C–N bond formation and its subsequent reduction. The protocol can be potentially applied in industrial settings and moreover, control of C–N bond formation processes with microwave, flow chemistry or rapid-heating approaches may facilitate selective establishment of production and synthesis methods for nitrogen-containing structures.

## Methods

**Reaction procedures**. A typical procedure for the microwave-assisted experiments was as follows. Reactants, furfural (FUR, 2 mmol) or other carbonyl compounds (1 or 2 mmol) and a given amount of formic acid (FA) and formamide (AM) or ammonium formate (AMF) were loaded into a thick-walled (wall thickness of 1.2 mm, inside diameter of 13.6 mm) Pyrex glass tube (10 mL, maximum pressure <10 MPa, Taiatsu Techno. Corp., Japan). The tube was then mounted into an outer

polycarbonate tube and sealed with two PEEK screw caps with special seal joints (Taiatsu Techno. Corp., Japan). A thermocouple was inserted into the reactor through a stainless steel sleeve (diameter in 1.5 mm), which was used to control the temperature inside the reactor. Prior to conducting reaction, N₂ gas was employed to purge air inside the reactor for 3 times and finally set to 2 MPa to suppress solution boiling. Vacuum was applied between the inner glass tube and the outer polycarbonate tube to reduce heat losses. All reactions were conducted using microwave heating apparatus (Shikoku Instrumentation Co. Ltd., Japan, model μ Reactor, SMW-087, 2.45 GHz, maximum power 700 W)[63–65]. Zero reaction time was defined as the point that the system attained the target reaction temperature. In the procedures, (12 to 40) s was required for heating with microwave irradiation from 20 °C to target temperatures of (120 to 200) °C. The specific reaction temperature could be nicely kept by automatically real-time change of microwave power. After passing the desired reaction time, tap water was introduced into the jacket of the tube (by internal reduced pressure provided by a vacuum pump) to rapidly cool its contents (Supplementary Fig. 44) and to terminate the reaction. The cooled reaction tube was depressurized and its contents were diluted with THF (2 mL) prior to sample analysis. Samples were analyzed with liquid chromatography using n-hexane/ethyl acetate (1:1, v/v) as mobile phase. Structures were confirmed by ¹H, ¹³C NMR (JEOL-ECX 500 NMR spectrometer, CDCl₃) and GC-MS (Supplementary Figs. 45–53). NMR spectra of some representative formamide products (**1** and **2**) are shown in Supplementary Figs 1 and 2.

In the oil-bath experiments, a Teflon-lined stainless steel autoclave was used (inner volume 10 mL), in which the target reaction temperatures were identical to those of the microwave-assisted experiments. Zero reaction time was defined as the point in which the autoclave was placed into an oil-bath that was preheated to 180 °C or another specific temperature. After a given reaction time, the autoclave was taken out of the oil bath and immediately cooled-down to ambient temperature with tap-water. After cooling, the autoclave was opened and THF was added to the reacted mixture that was analyzed. Separate experiments were conducted and repeated 2–3 times for each sample. The obtained conversions and yields are average of 2–3 individual experiments, with standard deviation ($\sigma$) in the range of 0.5–4.3%, as given by error bars in appropriate figures.

**Isotope-labeled experiments**. For the isotope-labeled studies, ¹H, ¹³C, and ¹H–¹³C HSQC NMR spectra of the reaction mixtures were used in the experiments at the given reaction conditions with either normal or deuterium reagents. Analyses were performed with deuterated solvents (THF-d₈ or CDCl₃) on a JEOL-ECX 500 NMR spectrometer. GC-MS spectra for some of the liquid mixtures were measured to determine the incorporated D in the products.

**Product analyses**. GC-MS (Agilent 6890N GC/5973 MS, Santa Clara, CA) was used to identify liquid products, intermediates and major by-products. GC-MS spectra of representative products are shown in Supplementary Fig. 40. Concentrations of organic acids (e.g., formic acid and levulinic acid) were determined with HPLC (LC-20A, Shimadzu, Kyoto) fitted with an Aminex HPX-87H column (Bio-Rad, Richmond, CA) and a refractive index (RI) detector. Quantitative analyses of the reaction mixtures were made with a GC (Agilent 7890B) with an HP-5 column (30 m × 0.320 mm × 0.25 μm) and a flame ionization detector with naphthalene as internal standard using standard curves (R² > 0.995) made from commercial samples or the isolated products. In addition, ¹H NMR was also used for

quantification of unpurified products in the reaction mixtures using 1,3,5-tri-methoxybenzene as the internal standard.

**Computational methods**. All DFT calculations were carried using the hybrid functional B3LYP[66] as implemented in Gaussian 09 D.01 software[67]. The all-electron 6-311+G(d,p) basis set was used for all atoms. The polarized continuum model (PCM)[68] with standard parameters for water solvent ($\varepsilon = 78.3$) was used to account for bulk solvent effects during geometry optimization, frequency analysis and searching of transition states. All relative energies discussed in this paper are referred to Gibbs energies considering the zero point energy (ZPE) correction at 453.15 K. For comparison, free energies were re-calculated at the level of M06-2X/def2-TZVP, and the relevant free energy diagrams are provided in supplementary information (Supplementary Fig. 54). After cross-checking, the B3LYP functional provides closer representation of the experimental data compared with M06-2X, so that the B3LYP functional was assessed to be suitable to simulate the studied system. In addition, corresponding 3D figures and Cartesian coordinates are supplied in Supplementary Figs. 55–58 and Supplementary Data 1.

## Data availability

The data that support the findings of this study are available from the corresponding authors upon reasonable request.

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

## Acknowledgements

This work is financially supported by Nanjing Agricultural University (68Q-0603), International Postdoctoral Exchange Fellowship Program of China (20170026), Post-doctoral Science Foundation of China (2016M600422), and Jiangsu Postdoctoral Research Funding Plan (1601029A) for study at Tohoku University. We thank Cheng-jiang Fang (Guizhou University) for recording part of the NMR spectra, and also acknowledge the Research Center of Supercritical Fluid Technology (Director, Prof. Hiroshi Inomata) of Tohoku University for logistics support. Special thanks to Professor Sir Martyn Poliakoff (University of Nottingham) for providing helpful comments on the manuscript.

## Author contributions

H.L., R.S., and Z.F. conceived the concept and directed the project. H.L., H.G., and Y.H. conducted experiments and analyses. Y.S. and E.J.M.H. performed the DFT calculations. H.L., R.S., M.W., and Z.F. discussed the experiments and results, and prepared the manuscript.

## Additional information

**Competing interests:** The authors declare no competing interests.

