## [Peer Review File · Nature Communications]

Reviewers' comments:

Reviewer #1 (Remarks to the Author):

The article reports a protocol to selectively produce formamides in one step and amines in two steps from biomass-derived ketones or aldehydes, with presence of formic acid and formamide or ammonium formate. Formic acid was claimed to perform as both H-donor and acid catalyst, and the former part was proved by control and isotope-labelling experiments. Kinetic studies were also provided to confirm the advantage of microwave irradiation. Condition optimization, reaction pathway elucidation and extension of substrate scope were also provided. This article is well prepared and organized, with majority of the claims proved with experimental evidence. However, I do not agree the acceptance of the paper in Nature Communication, mainly because neither the level of innovation nor the general applicability/wide implication of the article does not quite meet the high standard of the journal.

The major merits of the paper:

- 1) Biomass derived starting materials to N-containing chemicals;
- 2) Process intensification using microwaves;
- 3) Using formic acid instead of H₂ as hydrogen donor;
- 4) A switchable synthetic route to other formamide or amine based on reagent used;
- 5) Fairly reasonable understanding of reaction pathway.

These merits certainly justify the innovation of the article in a more specific journal, such as Green Chemistry, probably even Angew Chem Int Ed, but not for NC, which often carry highly innovative pathway, or fundamentally new mechanisms. For instance, the amination of furfural and HMF by NH₃ (which is more difficult) has been reported by Hara group in TIT.

Apart from that, there are some minor issues that may be addressed by the authors before submitting to another journal:

- 1) N,N'-(furan-2-ylmethylene)diformamide (FDFAM) was regarded as a key intermediate in the reaction system, and its yields were presented in many places of discussion part. However, the structure confirmation of FDFAM was never provided both in manuscript and supporting information. GC-MS or NMR spectra should be provided.
- 2) Formic acid was claimed to be both H donor and acid catalyst. The later role should be further discussed and justified. For example, without formic acid, will the formation of 2 be affected from the reaction between 1 and FUR?
- 3) Page 4, Fig. 1, what is "HCOOR" representing? Clearly justification is necessary.

Reviewer #3 (Remarks to the Author):

In this work, a protocol is developed that shows in situ formed N-formyl quasi-catalytic species afford highly selective synthesis of formamides or amines with controllable levels from aldehyde- and ketone- derived platform chemical substrates under solvent-free conditions. Some interesting results were obtained, but the manuscript need major revisions before possible publication. The comments are as follows:

1. The authors used furfural as reactant. Can this protocol be used to other aldehydes and ketones? The authors should include more experiments to show the versatility of this method and compare the advantages and disadvantages with previous methods.

2. In Section "Mode of heating", the authors reported that compared with the microwave-assisted process at 180 °C for complete conversion of furfural (37, 133 min; Table 1, entry 4), only 83% furfural conversion was attained with oil-bath heating for 60 min (Table 1, entry 5). The authors stated that microwave facilitates C-N bond formation between furfural and formamide or product 1 that contains the amino group. The effect of microwave irradiation is very critical for the results in this work. Although experimental results were listed in Table 1, the underlying mechanism was not discussed. It is needed to give deep understating of the underlying mechanism for the high-quality journal of Nat. Commun. The underlying mechanism is also important for the effective use of microwave irradiation in an apparatus of larger scale.

3. With the addition of 5 wt% water, the effect on the product distribution is significant, and the carbon balanced decrease to 88-89%. What is the carbon balance with addition of 25 wt% water in Figure S17? What are the products and what are the side reactions?

4. After passing the desired reaction time, tap water was introduced into a jacket of the tube to rapidly cool its contents and to terminate the reaction. How long is the time required to cool the reactant mixture. This may have effect on the results because the reaction time is quite short, several minutes.

5. Zero reaction time was defined as the point that the system attained the target reaction temperature. How was the reaction temperature kept at the target value? At a specific power of microwave irradiation, how did the reaction temperature change during the reaction process?

6. It is unclear if the authors sampled the reaction mixture during the reaction or if the data at each time point in Figs. 2(a) and 2(b) were obtained by a separate experiment? The author should clarify this point.

7. The reproducibility of the result should be addressed. Error bar should be included for important results, such as Fig. 2. The reproducibility should be checked by repeating the important experiments.

Response to the Reviewers' Comments:

Reviewer #1 (Remarks to the Author):

The article reports a protocol to selectively produce formamides in one step and amines in two steps from biomass-derived ketones or aldehydes, with presence of formic acid and formamide or ammonium formate. Formic acid was claimed to perform as both H-donor and acid catalyst, and the former part was proved by control and isotope-labelling experiments. Kinetic studies were also provided to confirm the advantage of microwave irradiation. Condition optimization, reaction pathway elucidation and extension of substrate scope were also provided. This article is well prepared and organized, with majority of the claims proved with experimental evidence. However, I do not agree the acceptance of the paper in Nature Communication, mainly because neither the level of innovation nor the general applicability/wide implication of the article does not quite meet the high standard of the journal.

The major merits of the paper:

- 1) Biomass derived starting materials to N-containing chemicals;
- 2) Process intensification using microwaves;
- 3) Using formic acid instead of H₂ as hydrogen donor;
- 4) A switchable synthetic route to other formamide or amine based on reagent used;
- 5) Fairly reasonable understanding of reaction pathway.

These merits certainly justify the innovation of the article in a more specific journal, such as Green Chemistry, probably even Angew Chem Int Ed, but not for NC, which often carry highly innovative pathway, or fundamentally new mechanisms. For instance, the amination of furfural and HMF by NH₃ (which is more difficult) has been reported by Hara group in TIT.

Response: The quasi-catalytic amination chemistry is demonstrated for furfural (**Image A**), in which the *N*-formyl species (formamide (AM), *N*-formyl imine, FDFAM) are found to not only sequentially activate elementary reaction steps of the amination, but also to stabilize the reaction species. FDFAM, which is a key species in the reaction chemistry, is quantifiable (GC-MS, Fig. S8), and when rapidly formed from the *N*-formyl imine in the early stages of the reaction, it allows unprecedented access to selective amination without catalyst and it is broadly applicable to not only biomass-derived carboxides (Table 2), but also to sugars or furfuryl aldehydes (new Table 3), carbonyl compounds containing carboxylic acids, aldehydes, and/or ketone groups (new Fig. 5), dicarbonyl compounds (new Fig. 6), and carboxylic acid – aldehyde or ketone ternary mixed-substrates (new Table 4).

To confirm the reaction mechanism, we added two collaborators (Su, Hensen, TU/e) and considered three different paths (new Fig. 4) with DFT calculations, whereas **Image A** is a subset. Amination proceeding *via N*-formyl imine preferably gives product **1** due to its relatively low overall reaction energy barrier between TS3* and *N*-formyl imine (ΔE : 88), while FDFAM with very low free energy (-211) is stably formed from *N*-formyl imine in the early stages of the reaction (main text, Fig. 2) to allow high carbon balance.

In previous reports of Hara's group (*J. Am. Chem. Soc.* (2017), *Chem. Sci.* (2018)), an imine that could not be detected was proposed to be a key intermediate to primary amines (**Image B**) via the reductive amination over metal catalysts.

In the absence of formic acid or a catalyst (**Image A**), the energy barrier for the initial formation of C-N bond using NH_3 (TS1# blue, 174 kJ mol^{-1}) is much higher than that for the present reaction system (TS1* red, 85 kJ mol^{-1}) with formamide (AM) under identical conditions, explicitly indicating that the *N*-formyl species (AM, *N*-formyl imine, FDFAM) have vital roles in the amination chemistry that are different from previously proposed amination chemistries.

The quasi-catalytic amination chemistry was applied to several different substrates (**Image-C**) to demonstrate its broadness and its easy access to aminated chemical products including ternary substrates to form combination compounds and dicarbonyl compounds to form heterocyclic compounds. Revised Fig. 1 (main text) highlights the broadness and versatility of the quasi-catalytic amination chemistry, while newly prepared tables and figures provide detailed and significant variation of the substrates.

Apart from that, there are some minor issues that may be addressed by the authors before submitting to another journal:

1) *N,N'*-(furan-2-ylmethylene)diformamide (FDFAM) was regarded as a key intermediate in the reaction system, and its yields were presented in many places

of discussion part. However, the structure confirmation of FDFAM was never provided both in manuscript and supporting information. GC-MS or NMR spectra should be provided.

Response: We are hopeful that the revised content will be of interest to readers of Nature Communications. The GC-MS of the FDFAM species was inadvertently left out of the initial submission and is now included in supporting information (Fig. S8).

2) Formic acid was claimed to be both H donor and acid catalyst. The later role should be further discussed and justified. For example, without formic acid, will the formation of 2 be affected from the reaction between 1 and FUR?

Response: Both H-donor and acidic roles of formic acid have been explicitly elucidated by the DFT calculations, and relevant results (Figs. 4, S32, and S33) and detailed discussion have been incorporated into the revised manuscript (p. 14-16).

The reaction suggested by the reviewer was also examined and <3% yield of **2** was obtained from **1** and FUR without FA under otherwise identical conditions, thus showing the consistency of the discussion on the proposed amination mechanism.

3) Page 4, Fig. 1, what is 'HCOOR' representing? Clearly justification is necessary.

Response: In the present study, HCOOR is limited to methyl formate, which has been justified in the main text as: "methyl formate (HCOOMe)".

Reviewer #3 (Remarks to the Author):

In this work, a protocol is developed that shows in situ formed N-formyl quasi-catalytic species afford highly selective synthesis of formamides or amines with controllable levels from aldehyde- and ketone- derived platform chemical substrates under solvent-free conditions. Some interesting results were obtained, but the manuscript need major revisions before possible publication. The comments are as follows:

1. The authors used furfural as reactant. Can this protocol be used to other aldehydes and ketones? The authors should include more experiments to show the versatility of this method and compare the advantages and disadvantages with previous methods.

Response: Besides furfural or even xylose as the starting substrate, the developed protocol is applicable to many other carbonyl compounds. Graphically, some of these are summarized by a revised Figure 1 (main text) and by **Image C** that provides an overview of the broad range of substrates in new figures and tables:

Amination of: (a) sugars (S1) and other nitrogen sources gives furfuryl amines (Table

3); (b) aldehydes and ketones (S2) gives *N*-formamides or primary amines (Table 2) (c) carboxylic acids (S3) gives *N*-formylimides (Fig. 5); (d) sterically-hindered dicarbonyl compounds (S4) gives polymer-free monomers (Fig. 5); (e) sterically-unhindered (spatial freedom) (S5) gives heterocyclic N-containing compounds (Fig. 6); (f) mixed-substrates or ternary substrates (S6) gives, most interestingly, combination compounds (Table 4).

2. In Section "Mode of heating", the authors reported that compared with the microwave-assisted process at 180 °C for complete conversion of furfural (3 min; Table 1, entry 4), only 83% furfural conversion was attained with oil-bath heating for 60 min (Table 1, entry 5). The authors stated that microwave facilitates C-N bond formation between furfural and formamide or product 1 that contains the amino group. The effect of microwave irradiation is very critical for the results in this work. Although experimental results were listed in Table 1, the underlying mechanism was not discussed. It is needed to give deep understating of the underlying mechanism for the high-quality journal of Nat. Commun. The underlying mechanism is also important for the effective use of microwave irradiation in an apparatus of larger scale.

Response: The reaction mechanism has been explored in detail. Please note that the FDFAM species GC-MS analyses were inadvertently omitted in the previously submitted manuscript. We explored possible reaction pathways with DFT calculations (Figures 4, S32, and S33) through collaboration with Hensen's group (TU/e) and found that experimental results are consistent with simulation results. In the amination process, *N*-formyl species together with formic acid as acid and H-donor was demonstrated to play a vital role in enhancing the selectivity and controllability of the reactions. The *in situ* formed *N*-formyl imine is more preferable to access the target product **1**, while FDFAM is stable and detectable by GC-MS, which is the key intermediate that provides high carbon balance by stabilizing the reaction species.

The initial C-N bond formation (Fig. 4, main text) was observed to be energy barrier (TS1, 79 kJ mol⁻¹) in the rate-determining reaction step that could be overcome by performing the amination at a relatively high temperature. In this case, rapid heating may exhibit a great potential in obtaining comparable performance to the system mediated by microwave irradiation and in future flow chemistry studies.

3. With the addition of 5 wt% water, the effect on the product distribution is significant, and the carbon balanced decrease to 88-89%. What is the carbon balance with addition of 25 wt% water in Figure S17? What are the products and what are the side reactions?

Response: When ammonium formate (AMF) is used instead of formamide, the addition of water promotes the formation of formic acid (FA) through hydrolysis. An appropriate amount of FA (as H-donor and acid) is necessary for the cascade reaction to proceed (hydrogenation and dehydration), while the presence of relatively high amounts of FA (as *N*-formyl stabilizing species) might inhibit the formation of tertiary amine **3** to yield more secondary and tertiary amides (**1** and **2**). However, it should be noted that the presence of excess water (e.g., 25 wt%) would result in the occurrence of side reactions such as furan-ring opening and condensation to give unknown byproducts or insoluble humins, thus depressing the carbon balance (ca. 86%). A sentence has been added into the revised manuscript (p. 11) to note the effect of excess water when using ammonium formate.

4. After passing the desired reaction time, tap water was introduced into a jacket of the tube to rapidly cool its contents and to terminate the reaction. How long is the time required to cool the reactant mixture. This may have effect on the results because the reaction time is quite short, several minutes.

Response: When the reaction system is at 180 °C, cooling to 100 °C is within 10 s to 15 s, and typically an additional 5 s to 10 s is required to reduce the temperature to 50 °C. At temperatures below 100 °C, no significant reaction takes place. The representative temperature and power profiles of the microwave reactor conducted at 180 °C in variable time are given in Fig. S37. This fast heating-up and rapid cooling-down system may also maintain the relatively high carbon balance and product selectivity for the amination process.

5. Zero reaction time was defined as the point that the system attained the target reaction temperature. How was the reaction temperature kept at the target value? At a specific power of microwave irradiation, how did the reaction temperature change during the reaction process?

Response: The heat losses during the reaction process are significantly reduced by an evacuated inner glass tube of the reactor. With real-time change of microwave power, a specific reaction temperature can be maintained without undue heat losses and without undue energy input. Representative profiles of temperature and power plotted with time for the microwave reactor are shown in the Fig. S37.

In addition, at a specific power of microwave irradiation, the reaction temperature was examined to initially increase and then keep in constant after a specific time (Figure S6, S7). More details on the microwave heating apparatus (Shikoku Keisoku, Japan, model μ Reactor, SMW-087, 2.45 GHz, maximum power 700 W) can be found in previously published papers (*Catal. Commun.* 2008, 9, 2244-2249; *Green Chem.* 2008, 10, 799-805; *ChemSusChem* 2010, 3, 1071-1077).

6. It is unclear if the authors sampled the reaction mixture during the reaction or if the data at each time point in Figs. 2(a) and 2(b) were obtained by a separate experiment? The author should clarify this point.

Response: All of the results obtained in this study were analyzed by separate experiments after the reaction was terminated by lowering the reactor temperature, and the experiments were repeated 2 to 3 times. Related clarifications have been included in the reaction procedures of Methods.

7. The reproducibility of the result should be addressed. Error bar should be included for important results, such as Fig. 2. The reproducibility should be checked by repeating the important experiments.

Response: The obtained conversions and yields (shown in tables and figures) are the average of 2 to 3 individual experiments, with standard deviation (σ) in the range of 0.5% to 4.3%. Error bars have been added to Fig. 2 as well as appropriate figures. A small section of text is added to the Methods section, p. 24.

Reviewers' comments:

Reviewer #1 (Remarks to the Author):

The major addition to the MS is DFT calculations which provide more insights into the reaction pathway and mechanism. The authors also make it clearer the uniqueness of the current paper by comparing in more detail previous works on making nitrogen-containing chemicals using metal-based catalysts. As such, I recommend the acceptance of the paper in its revised form.

Reviewer #3 (Remarks to the Author):

The authors adequately addressed the issues I raised, except Comment #2 concerning the effect of microwave heating. The authors did not directly answer my question. With the microwave-assisted process at 180 °C, complete conversion of furfural (3 min; Table 1, entry 4) was achieved, while only 83% furfural conversion was attained with oil-bath heating for 60 min (Table 1, entry 5). The authors stated that microwave facilitates C-N bond formation between furfural and formamide or product 1 that contains the amino group. It is needed to clearly discuss the underlying mechanism why microwave heating significantly enhanced the reaction rate.

Reviewer #4 (Remarks to the Author):

First of all, as a reviewer of the DFT part, I admit that I read the experimental discussion in less detail than the computations. As chemist involved in industrial projects related to amination I do note though, that impressive selectivities and yields can be reached for a wide range of substrates to this is certainly remarkable work. I rely on the expertise of the previous reviewers to assess this part and its novelty and related to this I have an important remark:

Indeed, I am surprised that a classical organic chemistry reaction such as the Leuckart synthesis, closely related to what is shown here, and known for more than a century, is not mentioned in the state of the art (see for example <https://pubs.acs.org/doi/abs/10.1021/jo01145a001>). Instead, only work of the last 10 years is cited. The current work should be placed in the full historical context of previous work and the novelty must be (re-)assessed.

To cite only the first paragraph of the JOC article: "Crossley and Moore (4) published a report of a comparison of the efficiency of the various reactants which may be used in the Leuckart reaction. They reported that formamide, ammonium formate, a mixture of formamide and formic acid, or a mixture of formamide and ammonium formate may be used with a carbonyl compound to give the reaction and that the mixture of formamide and formic acid produces the best yields."

Comments on the calculations:

The graphical representation of the reaction mechanism (Fig. 4) needs to be improved to match the quality of the other figures. In particular, some of the molecular structures are impossible to decipher. Since we are dealing with organic chemistry, it may be advisable to use a chemdraw / arrow-pushing representation and use the 3D representation only for key intermediates, or to provide large 3D figures in the SI with bond distances and so on. Further, the energy scheme is not drawn to scale and hence misleading (compare intermediate FDFAM to the rest, and for example the barrier between TS 6 and FDFAM). Finally, 1 should be fully drawn out in the scheme in the lower left.

The authors need to provide Cartesian coordinates of all intermediates in the supporting information, as it is common practice.

From a methodological point of view, the free energies should be computed for the reaction T (180 C?) and not for room temperature. This may have a significant impact on the energy landscape for

some of the steps. I guess that the pressure is not 1 bar but higher and this should also be accounted for in the free energies. Also, standard state corrections (see for example <http://www.ccl.net/chemistry/resources/messages/2014/05/01.004-dir/> and literature about solvation models) should be included if possible. While B3LYP is a common choice for organic chemistry, I would advise the authors to recalculate the electronic energies, at least at the single point level, with more modern functionals (at least something like M06-2X, wB97x-D ...) and with a better basis set (def2-TZVP or the like) and to provide this information in article text or SI. Depending on the computational resources of the authors, correlated wavefunction or composite methods (CBS-QB3) could be feasible for some of the steps at least to benchmark the DFT method, although I do not think that this is the priority here but rather see my comments below.

In the text, several times electronic energies are discussed ("Delta E"), although I believe free energies are meant.

My main concern are the assumptions of the mechanism itself. First, the starting point for the calculations are not furfural and AM, as stated in the text, but furfural and the imine of FA (formamidic acid, see graphical representation of FU + AM as starting point of the mechanism in Fig 4.). Unless there is a good reason for choosing this higher-energy tautomer (I may have misunderstood a key point and in this case I apologize), at least the first step of the mechanism (the formation of the carbinol species) is invalid and needs to be recalculate or the tautomerization step needs to be computed.

Second, it is not clear whether the energies have been referenced to separated furfural and the imine or the adduct between those molecules. In the caption it says furfural but the zero point is represented as the adduct. The correct way would be to reference to separated molecules (furfural, AM, FA). I doubt that at 180 C, furfural and AM form an adduct stable in free energy vs reactants.

In the text it is mentioned that high reaction T is necessary to overcome a barrier of ~80 kJ mol (TS1). Such a barrier is crossed at room temperature ! It is certainly not necessary to heat to 180 C, which should already have given the authors a hint that their propose mechanism cannot be correct.

It is claimed that pathway b is the most likely, but pathway c is thermodynamically most favorable. To conclude, it is necessary to compute the barrier from N-formylimine + AM to FDFAM and compare to the rest.

Other questions:

Is CO₂ release observed in the experiments?

What is the structure HCO₂* ? Formiate? Formyl radical?

What is the meaning of "hydrogenated dehydration"?

How does one model microwave effects in a DFT calculation?

Comments on other parts

The abstract is very hard to understand, for example "afford highly selective synthesis of formamides or amines with controllable levels of amination". What is meant by synthesis of amines with controllable level of amination?

Line 27: kinetic, control ?

Conclusions

I find significant flaws with the computations and cannot recommend publication without full revision.

The experimental work should be placed in the full context of the Leuckart and related synthesis and its novelty reassessed, from the point of view of the chemical route, yields, selectivities and

mechanistic insight. Unless this is properly done and clear differentiation to the state of the art is demonstrated, I cannot recommend publication in a top journal such as Nature communications.

Response to Reviewers' Comments:

Reviewers' comments:

Reviewer #1 (Remarks to the Author):

The major addition to the MS is DFT calculations which provide more insights into the reaction pathway and mechanism. The authors also make it clearer the uniqueness of the current paper by comparing in more detail previous works on making nitrogen-containing chemicals using metal-based catalysts. As such, I recommend the acceptance of the paper in its revised form.

Response: The authors greatly appreciate your positive evaluation and recommendation.

Reviewer #3 (Remarks to the Author):

The authors adequately addressed the issues I raised, except Comment #2 concerning the effect of microwave heating. The authors did not directly answer my question. With the microwave-assisted process at 180 °C, complete conversion of furfural (3 min; Table 1, entry 4) was achieved, while only 83% furfural conversion was attained with oil-bath heating for 60 min (Table 1, entry 5). The authors stated that microwave facilitates C-N bond formation between furfural and formamide or product 1 that contains the amino group. It is needed to clearly discuss the underlying mechanism why microwave heating significantly enhanced the reaction rate.

Response: The effect of microwave heating in the C-N bond formation (first elemental step) between furfural (FUR) and formamide (AM) can be explicitly addressed by the following points that are related to the heating rate (kinetics, activation energy) and that include an additional supplementary experiment that uses SiC particles that completely absorb microwave energy (pronounced convective heating):

(1) Under oil-bath heating conditions without microwave heating, the experimental activation energy is found to be 148 kJ mol⁻¹, which is in good agreement with the DFT-predicted [T = 180 °C, B3LYP/6-311+G(2s,2p)] value of 152 kJ mol⁻¹ in the presence of formic acid (FA); the activation energy is lowered to ca. 50 kJ mol⁻¹ with microwave heating, clearly indicating the promotional role that microwave heating has on lowering the reaction barrier toward establishing the C-N bond.

(2) Without the assistance of formic acid (FA), the DFT-predicted activation energy is 245 kJ mol⁻¹, which is much higher than values obtained with FA under either oil heating (experimental: 148 kJ mol⁻¹; theory: 152 kJ mol⁻¹) or microwave heating (experimental: 50 kJ mol⁻¹).

(3) Significantly inferior reactivity between **1** and FUR to **2** (ca. 15% yield) was observed in the presence of Et₃N equivalent to FA (in contrast, 83% yield of **2** was obtained without adding Et₃N), showing the significant role of FA acting as an acid. These results show that the dissociation of FA to give more free acidic species with enhanced acidity under microwave heating most likely accelerates the process of C-N bond formation. This speculation is also well confirmed by the wide application of the developed catalytic system in the context (e.g., Table 3, Figure 6) for the acid-mediated reaction steps such as dehydration and cyclization.

(4) With the addition of SiC powder into a reaction mixture of FUR, AM, and FA (Table S4), the reaction rate can be further accelerated (reaction completion within ca. 2.5 min, shorter than 3 min) despite forming a certain amount of carbon, which also affirms the positive role of microwave heating in the reaction process. Relevant discussion has been included in the revised manuscript and supporting information.

Reviewer #4 (Remarks to the Author):

First of all, as a reviewer of the DFT part, I admit that I read the experimental discussion in less detail than the computations. As a chemist involved in industrial projects related to amination I do note though, that impressive selectivities and yields can be reached for a wide range of substrates to this is certainly remarkable work. I rely on the expertise of the previous reviewers to assess this part and its novelty and related to this I have an important remark:

Indeed, I am surprised that a classical organic chemistry reaction such as the Leuckart synthesis, closely related to what is shown here, and known for more than a century, is not mentioned in the state of the art (see for example <https://pubs.acs.org/doi/abs/10.1021/jo01145a001>). Instead, only work of the last 10 years is cited. The current work should be placed in the full historical context of previous work and the novelty must be (re-)assessed.

To cite only the first paragraph of the JOC article: “Crossley and Moore (4) published a report of a comparison of the efficiency of the various reactants which may be used in the Leuckart reaction. They reported that formamide, ammonium formate, a mixture of formamide and formic acid, or a mixture of formamide and ammonium formate may be used with a carbonyl compound to give the reaction and that the mixture of formamide and formic acid produces the best yields.”

Response: Researchers performing aminations are widely aware of Leuckart syntheses, Leuckart-Wallach syntheses and their many variations as a classical approach for producing *N*-formylated compounds in moderate selectivity. Amination levels are generally uncontrollable and long reaction times are used thus the key mechanisms are overlooked. Even in microwave studies, e.g. ref. [1], the importance of short reaction times with fast-heating rates and the use of formic acid to generate the quasi-catalytic species (and intermediate) are overlooked and are completely missed. In the previous revision, we refrained from repeating the results or chemistry, since the reaction pathways and mechanisms are based on long reaction times. However, for clarification of the difference between Leuckart-type reaction

and our present work, several sentences are added for relevant literature (numbered as 42, 43, and 44) with corresponding discussion (Page 4), and the novelty of the present work is conveyed accordingly.

[1] Loupy, A., Monteux, D., Petit, A., Aizpurua, J. M., Domínguez, E., Palomo, C. Towards the rehabilitation of the Leuckart reductive amination reaction using microwave technology. *Tetrahedron Lett.* 37, 8177-8180 (1996).

Comments on the calculations:

The graphical representation of the reaction mechanism (Fig. 4) needs to be improved to match the quality of the other figures. In particular, some of the molecular structures are impossible to decipher. Since we are dealing with organic chemistry, it may be advisable to use a chemdraw / arrow-pushing representation and use the 3D representation only for key intermediates, or to provide large 3D figures in the SI with bond distances and so on. Further, the energy scheme is not drawn to scale and hence misleading (compare intermediate FDFAM to the rest, and for example the barrier between TS 6 and FDFAM). Finally, 1 should be fully drawn out in the scheme in the lower left.

Response: Fig. 4 has been updated according to reviewer suggestions (in chemdraw/arrow-pushing style with 3D representation only for key intermediates and transition states), and the relevant 3D figures have been supplied in supplementary information (Figs. S40 & S41).

The authors need to provide Cartesian coordinates of all intermediates in the supporting information, as it is common practice.

Response: Corresponding Cartesian coordinates have been provided in the SI (Fig. S42).

From a methodological point of view, the free energies should be computed for the reaction T (180 C?) and not for room temperature. This may have a significant impact on the energy landscape for some of the steps. I guess that the

pressure is not 1 bar but higher and this should also be accounted for in the free energies. Also, standard state corrections (see for example <http://www.ccl.net/chemistry/resources/messages/2014/05/01.004-diff/> and literature about solvation models) should be included if possible. While B3LYP is a common choice for organic chemistry, I would advise the authors to recalculate the electronic energies, at least at the single point level, with more modern functionals (at least something like M06-2X, wB97x-D ...) and with a better basis set (def2-TZVP or the like) and to provide this information in article text or SI. Depending on the computational resources of the authors, correlated wavefunction or composite methods (CBS-QB3) could be feasible for some of the steps at least to benchmark the DFT method, although I do not think that this is the priority here but rather see my comments below.

Response: Fig. 4 has been updated by re-computing the free energies at the reaction temperature of 180 °C. We considered the standard state corrections for the free energies. B3LYP is indeed a common choice for these type of calculations. For comparison, we also calculated the free energies at the level of M06-2X/def2-TZVP, and the relevant free energy diagram is provided in SI (Fig. S39). In comparison with the experimental activation energy (148 kJ mol⁻¹), the B3LYP functional does a better job than the M06-2X one (152 vs. 104 kJ mol⁻¹). Therefore, we think that the B3LYP functional is appropriate to simulate the studied catalytic system.

In the text, several times electronic energies are discussed (“Delta E”), although I believe free energies are meant.

Response: Yes, that is correct. The terminology “Delta G” for the free energy was unified and corrected in the revised manuscript.

My main concern are the assumptions of the mechanism itself. First, the starting point for the calculations are not furfural and AM, as stated in the text, but furfural and the imine of FA (formamidic acid, see graphical representation of FU + AM as starting point of the mechanism in Fig 4.). Unless there is a good

reason for choosing this higher-energy tautomer (I may have misunderstood a key point and in this case I apologize), at least the first step of the mechanism (the formation of the carbinol species) is invalid and needs to be recalculate or the tautomerization step needs to be computed.

Response: That is correct and it is a better starting position to demonstrate our point. In the revised Fig. 4, the stable configuration of AM was used, and the mechanism of the first step was re-considered. Now, the predicted activation energy is calculated to be 152 kJ mol^{-1} , in accordance with oil-heating experimental activation energy (148 kJ mol^{-1}). To allow a simple check, corresponding 3D figures and Cartesian coordinates are supplied in Figs. S40-S42.

Second, it is not clear whether the energies have been referenced to separated furfural and the imine or the adduct between those molecules. In the caption it says furfural but the zero point is represented as the adduct. The correct way would be to reference to separated molecules (furfural, AM, FA). I doubt that at 180 C, furfural and AM form an adduct stable in free energy vs reactants.

Response: The free energy diagrams in the revised Fig. 4 and SI have been re-constructed by reference to the energies of the separate molecules (furfural, AM and FA). At $180 \text{ }^{\circ}\text{C}$, the adduct formation of furfural and AM is endergonic ($\Delta G = 47 \text{ kJ mol}^{-1}$), indicative of its lower stability compared with reactants and final products.

In the text it is mentioned that high reaction T is necessary to overcome a barrier of $\sim 80 \text{ kJ mol}$ (TS1). Such a barrier is crossed at room temperature! It is certainly not necessary to heat to 180 C, which should already have given the authors a hint that their propose mechanism cannot be correct.

Response: As discussed above, the re-predicted barrier was found to be 152 kJ mol^{-1} , as illustrated in the revised Figure 4 (as well as corresponding 3D figures and Cartesian coordinates supplied in Figures S40-S42), which is in accordance with oil-heating experimental activation energy (148 kJ mol^{-1}). Some confusion may have resulted from the choice of the meta-stable imine of AM that was previously

adopted for calculation, which accounted for the low energy barrier.

It is claimed that pathway b is the most likely, but pathway c is thermodynamically most favorable. To conclude, it is necessary to compute the barrier from N-formylimine + AM to FDFAM and compare to the rest.

Response: Free energies were re-calculated for this reaction step, and the reaction barrier was found to be 104 kJ mol^{-1} with the assistance of formic acid via **TS4**, which is higher than that for *N*-formylimine formation ($\Delta G = 72 \text{ kJ mol}^{-1}$) and further hydrogenation to the product **1** ($\Delta G = 62 \text{ kJ mol}^{-1}$). In addition, FDFAM with relatively low activation energy (15 kJ mol^{-1}) is difficult to be hydrogenated into product **1** ($\Delta G = 113 \text{ kJ mol}^{-1}$), rendering it to exist in a certain quantity during the initial reaction stage. Therefore, *N*-formyl imine seems to be involved in the predominant reaction route, while FDFAM as the key intermediate is most likely to act to sustain high carbon balances observed experimentally by stabilizing the reaction species.

Other questions:

Is CO₂ release observed in the experiments?

Response: Yes, CO₂ was detected by GC.

What is the structure HCO₂*? Formiate? Formyl radical?

Response: It is formic acid anion (HCOO⁻), as demonstrated by NBO analysis (Supplementary Figure S41).

What is the meaning of “hydrogenated dehydration”?

Response: This term was revised to “hydrogenolysis”.

How does one model microwave effects in a DFT calculation?

Response: Simulation of mechanistic microwave effects with DFT calculations is

a research topic. The DFT calculations were used to compare with the oil-heating results, which were demonstrated to agree well with each other. As proposed, microwave heating increases the number of activated intermediates in the reaction to further lower the required activation energy, thus facilitating the reaction occurrence. For example, the initial C-N bond formation was observed to be the rate-determining reaction step (Fig. 4), while the medium reaction energy barrier (152 kJ mol^{-1}) could be overcome by performing the amination at relatively high temperatures, which is in accordance with the oil-heating experimental activation energy (148 kJ mol^{-1}). However, under microwave irradiation, the experimental activation energy was found to be ca. 50 kJ mol^{-1} , clearly showing the promotional role of microwave in the facilitation of C-N bond formation.

Comments on other parts

The abstract is very hard to understand, for example “afford highly selective synthesis of formamides or amines with controllable levels of amination”. What is meant by synthesis of amines with controllable level of amination?

Line 27: kinetic, control?

Response: The expression “...formamides or amines with controllable levels from...” means that primary, secondary or tertiary formamides or amines may be selectively synthesized from a wide variety of substrates and that the desired result can be controlled. Both substrate (i.e. formic acid) and heating rate (kinetics) are important in generating the quasi-catalytic species that allow control of the amination (controllable level of amination).

For better understanding, the phrase “kinetic, control...” in previous Line 27 was adjusted in the slightly revised abstract.

Conclusions

I find significant flaws with the computations and cannot recommend

publication without full revision.

The experimental work should be placed in the full context of the Leuckart and related synthesis and its novelty reassessed, from the point of view of the chemical route, yields, selectivities and mechanistic insight. Unless this is properly done and clear differentiation to the state of the art is demonstrated, I cannot recommend publication in a top journal such as Nature communications.

Response: Computations have been carefully checked and re-calculated according to the basis required by the reviewer. Updated results are provided in Figures 4 and S39-S42 including relevant discussion. The difference between Leuckart-type reactions and this study are clarified to clearly convey the novelty of the present work in the revised manuscript.

REVIEWERS' COMMENTS:

Reviewer #3 (Remarks to the Author):

The authors have adequately addressed the issue I raised. I recommend the manuscript for publication in Nature Communications.

Reviewer #4 (Remarks to the Author):

I apologize for the delayed answer. I am satisfied with the replies of the authors and the modification of the manuscript. Rewriting the DFT part was worth the effort. However, there are still some (minor) unclear points I would like the authors to address before the manuscript can be published.

Below are my detailed comments / questions:

Leuckart reaction:

Thank you for providing this additional context which clarifies the novelty of the work.

Response: Fig. 4 has been updated according to reviewer suggestions ...

 comment / question:

Thank you for providing the new figure which is much easier to read, even if I prefer the energy diagram representation of the first version which makes it easier to see the connections between intermediates.

To what does the 2nd panel correspond in Fig S40? The (high E) pathway was discussed in the old version of the article but it was removed here. What is the reason?

Response: Fig. 4 has been updated by re-computing the free energies at the reaction temperature of 180 °C ...

 comment / question:

Thank you for providing these additional calculations and referencing to 180 C. However, I do not understand the difference between Fig S39 and Fig S40 (M06-2X). In S39, 104 kJ mol⁻¹ are reported, while in S40 it is 140 kJ mol⁻¹, i.e. much closer to the B3LYP result (and experiment). The energies of other points are also off. This needs to be clarified.

Also, how was the experimental activation E determined (for oilbath and MW experiments)? The corresponding Arrhenius plots should be provided.

Response: Free energies were re-calculated for this reaction step, and the reaction barrier was found to be 104 kJ mol⁻¹ ...

 comment / question:

To which step do the 15 kJ mol⁻¹ correspond? Removal of FA? What is the corresponding TS? I do not see it in S40. The sentence should be rewritten to explain step by step what happens. Also "renders it to exist" should be something like "accounts for the observation of XXX".

Response: It is formic acid anion (HCOO⁻), as demonstrated by NBO analysis (Supplementary Figure S41).

 comment / question:

OK. This confirms what I thought although from writing HCO₂* it was not clear.

Line 87-90.

The presence of the formyl species (N-formyl carbinolamine, N-formyl imine, and N,N'-(furan-2-ylmethylene)diformamide (FDFAM)), originating from the AM/FA/AMF reactants, is a key species in forming nitrogen-containing products at specific levels.

 comment / question:

Sentence unclear. What is the key species?

Line 331-333

"In view of the relatively low reaction barrier ($\Delta G = 62 \text{ kJ mol}^{-1}$ vs 113 kJ mol^{-1} , Fig. 4), product 1 is preferably formed from N-formyl imine than from FDFAM via TS5."

 comment / question:

If I understand well, the more favourable pathway goes through IM1 and the higher E one goes through the N-formyl imine. Here the opposite is said.

Response to Reviewers' Comments:

REVIEWERS' COMMENTS:

Reviewer #3 (Remarks to the Author):

1. The authors have adequately addressed the issue I raised. I recommend the manuscript for publication in Nature Communications.

Answer: The authors greatly appreciate your positive evaluation and recommendation.

Reviewer #4 (Remarks to the Author):

2. I apologize for the delayed answer. I am satisfied with the replies of the authors and the modification of the manuscript. Rewriting the DFT part was worth the effort. However, there are still some (minor) unclear points I would like the authors to address before the manuscript can be published.

Answer: Many thanks for your valuable comments to further improve our manuscript. The issues listed below have been clearly clarified with relevant modifications in the revised manuscript.

3. Below are my detailed comments / questions:

Leuckart reaction:

Thank you for providing this additional context which clarifies the novelty of the work.

Answer: Thank you for your positive comment.

4. Response: Fig. 4 has been updated according to reviewer suggestions ...

 comment / question:

Thank you for providing the new figure which is much easier to read, even if I prefer the energy diagram representation of the first version which makes it easier to see the connections between intermediates.

Answer: We agree that the diagram is easier to visualize although it would occupy a lot of space if drawn to scale. However, we added the diagram in SI as Supplementary Figure 33 and note that the plotted values are not to scale as some readers may prefer this form.

To what does the 2nd panel correspond in Fig S40? The (high E) pathway was discussed in the old version of the article but it was removed here. What is the

reason?

Answer: The high energy pathway was found to be irrelevant to the analysis so that it was removed to avoid unnecessary discussion.

5. Response: Fig. 4 has been updated by re-computing the free energies at the reaction temperature of 180 °C ...

 comment / question:

Thank you for providing these additional calculations and referencing to 180 C. However, I do not understand the difference between Fig S39 and Fig S40 (M06-2X). In S39, 104 kJ mol⁻¹ are reported, while in S40 it is 140 kJ mol⁻¹, i.e. much closer to the B3LYP result (and experiment). The energies of other points are also off. This needs to be clarified.

Answer: The values are all consistent and give the same transition energies for a given basis set, namely, $E_a = \Delta E = E(\text{final}) - E(\text{initial})$. Initial energies of the starting materials depend on initial configurations and are not necessarily equal to zero as assumed by the reviewer.

For (Fig. S39 or Suppl. Fig. 41):

$E(\text{FUR} + \text{AM} + \text{FA}) = 0 \text{ kJ/mol}$, $E(\text{TS1}) = 104 \text{ kJ/mol}$, $E_a(\text{TS1}) = 104 - 0 = \underline{104 \text{ kJ/mol}}$

For (Fig. S40 or Panel D Suppl. Fig. 42):

$E(\text{FUR} + \text{AM} + \text{FA}) = 36 \text{ kJ/mol}$, $E(\text{TS1}) = 140 \text{ kJ/mol}$, $E_a(\text{TS1}) = 140 - 36 = \underline{104 \text{ kJ/mol}}$

To alert the reader to the relative values, a note has been added in the captions:

"Values in parentheses are free energies (kJ mol⁻¹) with respect to the starting energy of the three separated reactants FUR, AM and FA." (or reactants as appropriate)

The B3LYP/6-311+G(2s,2p) basis set provides activation energies closest to experiment and so these were used in analyses in main text.

6. Also, how was the experimental activation E determined (for oil bath and MW experiments)? The corresponding Arrhenius plots should be provided.

Answer: The experimental activation energies for MW and oil bath experiments were determined based on the FUR conversion plotted with reaction time (assumed as 1st order reaction). Corresponding Arrhenius plots are provided in Supplementary Figure 34.

7. Response: Free energies were re-calculated for this reaction step, and the reaction barrier was found to be 104 kJ mol⁻¹ ...

 comment / question:

To which step do the 15 kJ mol⁻¹ correspond? Removal of FA? What is the corresponding TS? I do not see it in S40. The sentence should be rewritten to explain step by step what happens. Also "renders it to exist" should be something like "accounts for the observation of XXX".

Answer: In Supplemental Figure 42 (Panel A and B), the free energy of 15 kJ

mol^{-1} ($17 - 2 = 15 \text{ kJ mol}^{-1}$, relative the starting material) corresponds to the sole FDFAM, other than together with FA, which was noted in Fig. 4. The transition state toward FDFAM was TS4 (with activation energy of $176 - 72 = 104 \text{ kJ mol}^{-1}$, relative to the former reactant), which could also be found in the original Figure S40 (renumbered as Supplemental Figure 42, Panel C), conformably marked as TS4, but with free energy of 178 kJ mol^{-1} , in view of the starting material bearing 2 kJ mol^{-1} free energy). The corresponding sentences have been revised as recommended (Page 15).

7. Response: It is formic acid anion (HCOO^-), as demonstrated by NBO analysis (Supplementary Figure S41).....

 comment / question:

OK. This confirms what I thought although from writing HCO_2^* it was not clear.

Answer: Acknowledged.

8. Line 87-90.

The presence of the formyl species (N-formyl carbinolamine, N-formyl imine, and N,N'-(furan-2-ylmethylene)diformamide (FDFAM)), originating from the AM/FA/AMF reactants, is a key species in forming nitrogen-containing products at specific levels.

 comment / question:

Sentence unclear. What is the key species?

Answer: The key species in the reaction mechanism is the in situ formed formyl groups, as mentioned in the first half sentence (presently, Page 4). The was rewritten to read as "Presence of the formyl species (N-formyl carbinolamine, N-formyl imine, and N,N'-(furan-2-ylmethylene)diformamide (FDFAM)), originating from AM/FA/AMF reactants, plays key roles in forming nitrogen-containing products at specific levels.". (Page 4)

9. Line 331-333.

"In view of the relatively low reaction barrier ($\Delta G = 62 \text{ kJ mol}^{-1}$ vs 113 kJ mol^{-1} , Fig. 4), product 1 is preferably formed from N-formyl imine than from FDFAM via TS5."

 comment / question:

If I understand well, the more favourable pathway goes through IM1 and the higher E one goes through the N-formyl imine. Here the opposite is said.

Answer: Yes, this was a writing error. The reaction proceeding through N-formyl carbinolamine (via IM1) rather than N-formyl imine should be the more favorable pathway toward product **1**. The incorrect word “N-formyl imine” has been adjusted to “N-formyl carbinolamine” (Page 15).